# TOWARDS SIMPLE AND PROVABLE PARAMETER-FREE ADAPTIVE GRADIENT METHODS

## ABSTRACT

Optimization algorithms such as AdaGrad and Adam have significantly advanced the training of deep models by dynamically adjusting the learning rate during the optimization process. However, adhoc tuning of learning rates poses a challenge, leading to inefficiencies in practice. To address this issue, recent research has focused on developing "learning-rate-free" or "parameter-free" algorithms that operate effectively without the need for learning rate tuning. Despite these efforts, existing parameter-free variants of AdaGrad and Adam tend to be overly complex and/or lack formal convergence guarantees. In this paper, we present AdaGrad++ and Adam++, novel and simple parameter-free variants of AdaGrad and Adam with convergence guarantees. We prove that AdaGrad++ achieves comparable convergence rates to AdaGrad in convex optimization without predefined learning rate assumptions. Similarly, Adam++ matches the convergence rate of Adam without relying on any conditions on the learning rates. Experimental results across various deep learning tasks validate the competitive performance of AdaGrad++ and Adam++.

## 1 INTRODUCTION

In recent years, optimization algorithms such as AdaGrad (Duchi et al., 2011) and Adam (Kingma, 2014) have emerged as powerful tools for enhancing the training of deep learning models by efficiently adapting the learning rate during the optimization process. While these algorithms have demonstrated remarkable performance gains in various applications, a notable drawback lies in the necessity of manual tuning for suitable learning rates. The process of learning rate tuning can be laborious and often requires extensive trial and error, hindering the efficiency and scalability of deep learning model development.

The intricate nature of learning rate tuning has motivated a large number of recent works to develop "learning-rate-free" or "parameter-free" algorithms that can work well under various different settings without learning rate tuning. Among the vast literature of parameter-free optimization methods, Ivgi et al. (2023) proposed a framework called distance over gradients (DoG), which gives a parameter-free version of stochastic gradient descent (SGD) that shares certain features as the AdaGrad-Norm algorithm (Streeter & McMahan, 2010; Ward et al., 2020). Motivated by AdaGrad-Norm, another recent work (Defazio & Mishchenko, 2023) also gave a framework named D-adaptation, and parameter-free variants of SGD and Adam were proposed under this framework. More recently, Defazio et al. (2024) proposed a different approach for schedule-free online optimization, based on which the authors developed new variants of schedule-free SGD and Adam/AdamW.

Despite the recent advances of parameter-free optimization algorithms, research on parameter-free adaptive gradient methods[1] remains relatively limited. Specifically, most of the existing parameter-free algorithms are essentially variants of SGD, and *entry-wisely adaptive learning rates* in standard AdaGrad and Adam algorithms are rarely considered in most of the existing parameter-free methods. Although Defazio & Mishchenko (2023); Mishchenko & Defazio (2023); Defazio et al. (2024)

---

[1]Adaptive gradient methods usually have multiple hyperparameters other than learning rates. For example, Adam implements exponential moving averages of first and second moments of gradients, which are controlled by parameters $\beta_1$ and $\beta_2$. Here we clarify that when discussing parameter-free adaptive gradient methods, we still allow the algorithm to have such hyperparameters which do not require extensive tuning. This is consistent with the convention in recent works on parameter-free optimization (Defazio & Mishchenko, 2023; Mishchenko & Defazio, 2023; Defazio et al., 2024).

recently proposed variants of parameter-free AdaGrad, Adam and AdamW that implement entry-wisely adaptive gradients, these algorithms all introduce rather significant modifications to the original algorithms, and the parameter-free versions of Adam/AdamW are not backed up by theoretical convergence guarantees.

Motivated by the limitations of existing studies, in this work, we propose simple but efficient versions of AdaGrad and Adam with provable convergence guarantees, which we name AdaGrad++ and Adam++ respectively. The main contributions of this work can be summarized as follows:

1. We propose the AdaGrad++ algorithm, which is a parameter-free version of AdaGrad. We demonstrate that without any assumptions on learning rates, AdaGrad++ can still achieve a $O(1/\sqrt{T})$ worst-case convergence rate in convex optimization, which is the same as AdaGrad. This highlights the efficacy and versatility of AdaGrad++ as a more accessible and user-friendly optimization method.

2. We also introduce the Adam++ algorithm as a parameter-free variant of Adam. By eliminating the reliance on a well-tuned learning rate schedule, Adam++ offers more enhanced adaptability and robustness compared to Adam. Our theoretical results demonstrates the capability of Adam++ to match the convergence rate of Adam in convex optimization, even in the absence of any assumptions regarding learning rates.

3. We conduct experiments on image classification and large language model pretraining tasks to evaluate the performance of the proposed algorithms. For CIFAR-10, with minimal parameter tuning, Adam++ outperforms Adam by 0.27% using a constant learning rate schedule on ResNet-50, and by 1.35% using a cosine learning rate schedule on VGG16. For GPT-2 small and medium tasks, AdamW++ surpasses Adam by 0.02 in both training and test losses. Additionally, we perform an ablation study on the choice of initial and base learning rates, which confirms our theoretical findings.

**Notation.** We denote scalars by lowercase letters, vectors by lowercase boldface letters, and matrices by uppercase boldface letters. For a positive integer $d$, we denote $[d] = \{1, \ldots, d\}$. For a vector $\mathbf{x} = [x_1, \ldots, x_d]^\top$ and $p \geq 1$, we denote the $\ell_p$ norm of $\mathbf{x}$ by $\|\mathbf{x}\|_p = \left( \sum_{i=1}^{d} |x_i|^p \right)^{1/p}$, and the $\ell_\infty$ norm of $\mathbf{x}$ by $\|\mathbf{x}\|_\infty = \max_{i \in [d]} |x_i|$. Given two sequences $\{a_n\}$ and $\{b_n\}$, we write $a_n = O(b_n)$ if there exists a constant $0 < C < +\infty$ such that $a_n \leq C b_n$. We use the notation $\widetilde{O}(\cdot)$ to hide logarithmic factors.

## 2 RELATED WORK

In this section, we give a more comprehensive review of the existing literature on parameter-free optimization and adaptive gradient methods.

**Parameter-free optimization.** Several recent works have explored parameter-free algorithms based on modifications of the Polyak step size (Loizou et al., 2021; Gower et al., 2021; Orvieto et al., 2022; Rolinek & Martius, 2018; Berrada et al., 2020). In addition, several studies have investigated step-size selection methods derived from Line-Search algorithms (Vaswani et al., 2019; Paquette & Scheinberg, 2018). Another line of works, including LARS (You et al., 2017a), LAMB (You et al., 2017b), Adafactor (Simonyan & Zisserman, 2014), and Fromage (Bernstein et al., 2020), introduced learning rate adjustment schemes based on the norms of iterates. Moreover, Chandra et al. (2022) proposed a scheme to adjust the learning rates based on certain automatically calculated hypergradients. Several recent works (Orabona & Tommasi, 2017; Chen et al., 2022) have also proposed parameter-free algorithms by reducing the optimization process to a game of betting on a coin. Another recent work (Kleinsorge et al., 2023) proposed a novel rotation invariant parameter-free algorithm based on exponential learning rate adaption. Finally, a line of recent works (Orabona, 2014; Kempka et al., 2019) have studied parameter-free algorithms in solving specific learning tasks such as linear and kernel regression.

**Adaptive gradient methods.** There is a large body of literature on variants of AdaGrad and Adam. Specifically, RMSProp (Kurbiel & Khaleghian, 2017) was the first work that proposed using an exponential moving average instead of a cumulative sum to handle the second moment in AdaGrad. Reddi et al. (2019) pointed out an extreme case where Adam may face convergence issues, and proposed AMSGrad accordingly with convergence guarantees. RMSProp, Adam and AMSGrad have also inspired many variants, including SC-AdaGrad, SC-RMSprop (Mukkamala & Hein,

2017), Sadagrad (Chen et al., 2018b), YOGI (Zaheer et al., 2018), Padam (Chen et al., 2018a), and RAdam (Liu et al., 2019). More recently, several works such as STORM (Cutkosky & Orabona, 2019), adaptive normalized SGD (Cutkosky & Mehta, 2020), Adam+ (Liu et al., 2020), SUPER-ADAM Huang et al. (2021) implemented various variance reduction techniques in Adam. Guo et al. (2021) presented a novel convergence analysis for a family of Adam-style methods with an increasing momentum parameter for the first-order moment. Alacaoglu et al. (2020) proposed a new type of framework to analyze the regret of the Adam style methods. Zhou et al. (2018) established high-probabiliy convergence guarantees of AdaGrad and Adam in nonconvex optimization.

## 3 REVIEW OF EXISTING METHODS AND PREVIEW OF PROPOSED METHODS

In this section, we give a brief review of the adaptive gradient methods, and discuss existing literature of parameter-free adaptive gradient methods, followed by a preview of our proposed methods.

We consider the optimization problem as follows

$$\min_{\mathbf{x} \in \mathbb{R}^d} f(\mathbf{x}), \tag{3.1}$$

where $f$ can be a convex or nonconvex function. In order to optimize (3.1), the standard stochastic gradient decent (SGD) performs the following update rule

$$\mathbf{x}_{t+1} = \mathbf{x}_t - \eta_t \mathbf{g}_t, \tag{3.2}$$

where $\mathbf{g}_t$ represents the stochastic gradient at the $t$-th iteration, $\eta_t$ denotes the learning rate. Adaptive gradient methods (Duchi et al., 2011; Hinton et al., 2012; Kingma, 2014; Reddi et al., 2018; Loshchilov & Hutter, 2019; Chen et al., 2020) aim to give well-designed adjustments to the learning rate $\eta_t$, particularly focusing on applying different learning rates for different entries of the iterates.

Among popular adaptive gradient methods, AdaGrad (Duchi et al., 2011) stands out as one of the pioneering methods. The update rule for AdaGrad is given by:

$$\mathbf{x}_{t+1} = \mathbf{x}_t - \frac{\eta_t}{\sqrt{\sum_{i=1}^{t} \mathbf{g}_i^2} + \delta} \cdot \mathbf{g}_t, \tag{3.3}$$

where $\delta$ is a small positive constant, and we use the common notation where the square $(\cdot)^2$ and square root $\sqrt{\cdot}$ operations are performed entry-wisely when applied to a vector.

Adam (Kingma, 2014) is another widely recognized adaptive gradient methods. Compared with AdaGard, it implements exponential moving averages over $\mathbf{g}_t^2$'s, as well as momentum acceleration, with the update rule defined as follows:

$$\mathbf{x}_{t+1} = \mathbf{x}_t - \eta_t \frac{\mathbf{m}_t}{\sqrt{\mathbf{v}_t} + \delta}, \quad \mathbf{m}_t = \beta_1 \mathbf{m}_{t-1} + (1 - \beta_1)\mathbf{g}_t, \quad \mathbf{v}_t = \beta_2 \mathbf{v}_{t-1} + (1 - \beta_2)\mathbf{g}_t^2. \tag{3.4}$$

Another line of research on parameter-free optimization seeks to reduce or remove the necessity of learning rate tuning. The distance over gradient (DoG) (Ivgi et al., 2023) framework is popular method which sets the learning rate $\eta_t$ in stochastic gradient descent (3.2) as

$$\eta_t = \frac{\max_{i \leq t} \|\mathbf{x}_0 - \mathbf{x}_i\|_2}{\sqrt{\sum_{i=1}^{t} \|\mathbf{g}_i\|_2^2}}.$$

DoG can be treated as a modification on the AdaGrad-Norm algorithm (Duchi et al., 2011; Streeter & McMahan, 2010; Ward et al., 2020) with $\eta_t = D/\sqrt{\sum_{i=1}^{t} \|\mathbf{g}_i\|_2^2}$, where the parameter $D$ is set as $\max_{i \leq t} \|\mathbf{x}_0 - \mathbf{x}_i\|_2$ in DoG. Several other parameter-free methods (Defazio & Mishchenko, 2023; Mishchenko & Defazio, 2023) also focused on estimating the parameter $D$ with different criteria. Notably, these recent studies of parameter-free algorithms focus more on the variants of SGD, which do not implement the entry-wisely adaptive learning rates in AdaGrad and Adam. Although several recent works (Defazio & Mishchenko, 2023; Mishchenko & Defazio, 2023; Defazio et al., 2024) proposed parameter-free variants of AdaGrad or Adam, they are mostly not backed up with theoretical guarantees. Moreover, existing parameter-free variants of AdaGrad and Adam are mostly pretty complicated, deviating significantly from the standard forms of AdaGrad and Adam.

**Preview of our proposed methods.** Inspired by DoG (Ivgi et al., 2023), we propose simple parameter-free variants of AdaGrad and Adam, which we call AdaGrad++ and Adam++ respectively. Specifically, AdaGrad++ follows the update rule of AdaGrad in (3.3), but with

$$\eta_t = d^{-1/2} \cdot \max_{i \leq t} \|\mathbf{x}_i - \mathbf{x}_0\|_2,$$

where $d$ is the dimension of $\mathbf{x}$. Note that $\eta_t$ is the maximum distance between the initialization $\mathbf{x}_0$ and all the iterates along the optimization trajectory normalized by $\sqrt{d}$. Moreover, a specific and simplified case in Adam++ is directly based on the update rule of Adam in (3.4), with

$$\eta_t = \frac{\max_{i \leq t} \|\mathbf{x}_i - \mathbf{x}_0\|_2}{\sqrt{d(t+1)}}.$$

Compared with existing parameter-free versions of AdaGrad and Adam, AdaGrad++ and Adam++ are in much simpler form. Interestingly, despite the simplicity, our analysis demonstrates that AdaGrad++ and Adam++ enjoy good theoretical convergence guarantees, and perform surprisingly well in various experiments. For more details, please refer to Sections 4 and 5.

## 4 ADAGRAD++: A PARAMETER-FREE VERSION OF ADAGRAD

In this section, we present the details of the AdaGrad++ algorithm, and then give theoretical guarantees on its performance in convex optimization.

### 4.1 ALGORITHM

We consider the optimization problem as introduced in (3.1) in setting of stochastic optimization, and we assume access to a *stochastic gradient oracle* $\mathcal{G}(\mathbf{x})$ satisfying $\mathbb{E}[\mathcal{G}(\mathbf{x})|\mathbf{x}] \in \partial f(\mathbf{x})$. The AdaGrad++ algorithm is presented in Algorithm 1.

---

**Algorithm 1** Parameter-Free AdaGrad (AdaGrad++)

---

1: **input:** $\mathbf{x}_0, \eta_0 = \epsilon, \delta$
2: **for** $t = 0$, **to** $n$ **do**
3:    $r_t = \|\mathbf{x}_t - \mathbf{x}_0\|_2/\sqrt{d}$
4:    $\eta_t = \max(\eta_{t-1}, r_t)$
5:    $\mathbf{g}_t = \mathcal{G}(\mathbf{x}_t)$
6:    $\mathbf{s}_t = (\sum_{k=0}^{t} \mathbf{g}_k^2)^{1/2}$
7:    $\mathbf{H}_t = \delta + \mathrm{diag}(\mathbf{s}_t)$
8:    $\mathbf{x}_{t+1} = \mathbf{x}_t - \eta_t \cdot \mathbf{H}_t^{-1}\mathbf{g}_t$
9: **end for**

---

In Algorithm 1, it is clear that the key innovation of AdaGrad++ lies in the introduction of the quantity $r_t = \|\mathbf{x}_t - \mathbf{x}_0\|_2/\sqrt{d}$, and the definition that $\eta_t = \max(\eta_{t-1}, r_t)$. These definitions are inspired by the DoG framework (Ivgi et al., 2023), and are the key to a parameter-free approach. We would also like to comment that introducing the factor $d^{-1/2}$ in the definition of $r_t$ is crucial in AdaGrad++, resulting in both strong theoretical guarantees and robust practical performance across different tasks with varying dimensions. The intuition is that AdaGrad++ implements different adaptive learning rates for different coordinates, and the $d^{-1/2}$ factor converts the "total distance" in DoG to the "mean squared distance (displacement)", which is more robust to $d$.

### 4.2 CONVERGENCE GUARANTEE

In this subsection, we present convergence guarantees of AdaGrad++ (Algorithm 1) under the setting where $f(\mathbf{x})$ is convex. We first give an assumption on the stochastic gradient $\mathcal{G}(\mathbf{x})$.

**Assumption 4.1.** There exists some continuous function $l : \mathbb{R}^d \to \mathbb{R}$ such that $\|\mathcal{G}(\mathbf{x})\|_2 \leq l(\mathbf{x})$ almost surely.

Assumption 4.1 states that the stochastic gradients have a deterministic bound $l(\mathbf{x})$ on their norm. By allowing different bounds at different $\mathbf{x}$, this assumption is much weaker compared to the more common Lipschitz assumption that directly requires that $\|\mathcal{G}(\mathbf{x})\|_2\|_2$ is bounded to a constant. The same assumption has been made in Ivgi et al. (2023).

Our main result on the convergence of AdaGrad++ is given in the following theorem.

**Theorem 4.2.** Let $\mathbf{x}_0, \ldots, \mathbf{x}_T$ be the iterates of AdaGrad++. Further let $\tau \in \arg\max_{t \leq T} \sum_{i=0}^{t-1} \frac{\eta_i}{\eta_t}$ and define $\overline{\mathbf{x}}_\tau = \frac{\sum_{t=0}^{\tau-1} \eta_t \mathbf{x}_t}{\sum_{t=0}^{\tau-1} \eta_t}$. Then under Assumption 4.1, for any $\delta \in (0, 1)$, $L > 0$ and any $\mathbf{x}_* \in \mathbb{R}^d$, with probability at least $1 - \delta - \mathbb{P}(\max_{t \leq T} l(\mathbf{x}_t) > L)$, it holds that

$$f(\overline{\mathbf{x}}_\tau) \leq f(\mathbf{x}^*) + O\left( \frac{D_\tau^2 \sqrt{d} \cdot \|\mathbf{s}_\tau\|_2 + \overline{D}_\tau \eta_0 \sqrt{\theta_{\tau,\delta} \|\mathbf{s}_\tau\|_2^2 + L^2 \theta_{\tau,\delta}^2}}{T\eta_0} \log\left( \frac{\eta_T}{\eta_0} \right) \right),$$

where $D_\tau = \max_{t \leq \tau} \|\mathbf{x}_t - \mathbf{x}^*\|_\infty$, $\overline{D}_\tau = \max_{t \leq \tau} \|\mathbf{x}_t - \mathbf{x}_*\|_2$, and $\theta_{t,\delta} = \log(\frac{60 \log(6t)}{\delta})$.

Theorem 4.2 gives the bound of $f(\overline{\mathbf{x}}_\tau)$ that is defined by an arbitrarily chosen reference point $\mathbf{x}_*$, and the bound contains a term $f(\mathbf{x}_*)$ as well as several other terms that are related to the distance between algorithm iterates and $\mathbf{x}_*$. This form of the bound with a reference point matches standard bounds in convex and Lipschitz/smooth optimization (Bubeck et al., 2015). Moreover, the probability for the bound in Theorem 4.2 to hold depends on $\mathbb{P}(\max_{t \leq T} l(\mathbf{x}_t) > L)$, and the bound holds with high probability when $\mathbb{P}(\max_{t \leq T} l(\mathbf{x}_t) > L)$ is small. It is clear that if $l(\cdot)$ is always bounded, which corresponds to a Lipschitz $f$, then $\mathbb{P}(\max_{t \leq T} l(\mathbf{x}_t) > L) = 0$ with an appropriately chosen constant $L$. In addition, it is also clear that Theorem 4.2 covers more general and non-Lipschitz cases as well, since $l(\cdot)$ only needs to be bounded along the optimization trajectory $\mathbf{x}_0, \ldots, \mathbf{x}_T$ to grant $\mathbb{P}(\max_{t \leq T} l(\mathbf{x}_t) > L) = 0$.

Theorem 4.2 reveals that an important term $\|\mathbf{s}_\tau\|_2$ determines the convergence rate of AdaGrad++. We note that a similar quantity has been investigated by Zhou et al. (2018) in the study of non-convex convergence guarantees of adaptive gradient methods. This similarity demonstrates that our proposed parameter-free algorithm AdaGrad++ still captures the key nature of AdaGrad. Taking a closer look at the quantity $\|\mathbf{s}_\tau\|_2$, by definition, we have $\|\mathbf{s}_\tau\|_2 = \sqrt{\sum_{t=0}^\tau \|\mathbf{g}_t\|_2^2}$. When the objective function is Lipschitz ($l(\cdot)$ is bounded), it is clear that a worst-case upper bound of $\|\mathbf{s}_\tau\|_2$ is $\sqrt{T}$, leading to a $1/\sqrt{T}$ bound on the convergence rate (see Corollary 4.3 below). However, as discussed in Zhou et al. (2018), here we point out that in practice, we often observe that $\|\mathbf{s}_\tau\|_2 \ll \sqrt{T}$ due to the fact that the algorithm converges and the stochastic gradients $\|\mathbf{g}_t\|_2$ may converge to zero. When $\|\mathbf{s}_\tau\|_2 = O(T^{1/2-\alpha})$ for some $\alpha \in (0, 1/2)$, we will have a better convergence rate of AdaGrad++ (see Corollary 4.4 below).

**Corollary 4.3.** Suppose that the assumptions in Theorem 4.2 hold. Further assume that $l(\mathbf{x}) \leq G$ for all $\mathbf{x}$. Then for any $\mathbf{x}^* \in \mathbb{R}^d$, with probability at least $1 - \delta$, it holds that

$$f(\overline{\mathbf{x}}_\tau) \leq f(\mathbf{x}^*) + \widetilde{O}\left( D_\tau^2 G \cdot \sqrt{\frac{d}{T}} \right),$$

where $D_\tau = \max_{t \leq \tau} \|\mathbf{x}_t - \mathbf{x}_*\|_\infty$.

Corollary 4.3 gives a simplified version of Theorem 4.2 under the special case when $l(\mathbf{x}) \leq G$. We note that Mishchenko & Defazio (2023) proposed a parameter-free version of AdaGrad named D-Adapted AdaGrad and established a convergence rate of the order $O(dG_\infty/\sqrt{T})$, under the assumption that $\|\mathcal{G}(\mathbf{x})\|_\infty \leq G_\infty$. Considering $\|\mathcal{G}(\mathbf{x})\|_2 \leq \sqrt{d} \cdot \|\mathcal{G}(\mathbf{x})\|_\infty$, we have $G \leq \sqrt{d} \cdot G_\infty$, and therefore our result can be reduced to the bound in Mishchenko & Defazio (2023) when we ignore the distance factor $D_\tau$.

**Corollary 4.4.** Suppose that the assumptions in Theorem 4.2 hold. Further assume that there exist $G > 0$ such that $l(\mathbf{x}) \leq G$ and $\|\mathbf{s}_\tau\|_2 \leq G \cdot T^{1/2-\alpha})$ for some $\alpha \in [0, 1/2)$. Then for any $\mathbf{x}^* \in \mathbb{R}^d$, with probability at least $1 - \delta$, it holds that

$$f(\overline{\mathbf{x}}_\tau) \leq f(\mathbf{x}_*) + \widetilde{O}\left( \frac{D_\tau^2 G \cdot \sqrt{d}}{T^{1/2+\alpha}} \right),$$

where $D_\tau = \max_{t \leq \tau} \|\mathbf{x}_t - \mathbf{x}_*\|_\infty$.

Corollary 4.4 is a straightforward simplification of Theorem 4.2 under the additional condition that $\|\mathbf{s}_\tau\|_2 \leq G \cdot T^{1/2-\alpha})$. It verifies that when the key quantity $\|\mathbf{s}_\tau\|_2$ is smaller than the worst-case $O(\sqrt{T})$ bound, the convergence rate can be faster than $O(1/\sqrt{T})$.

## 5 ADAM++: A PARAMETER-FREE VERSION OF ADAM

In this section, we introduce the Adam++ algorithm together with its theoretical convergence guarantees.

### 5.1 ALGORITHM

We consider the same optimization problem as introduced in (3.1) in the stochastic setting. We also consider the same stochastic gradient oracle $\mathcal{G}(\mathbf{x})$ satisfying $\mathbb{E}[\mathcal{G}(\mathbf{x})|\mathbf{x}] \in \partial f(\mathbf{x})$. The Adam++ algorithm is depicted in Algorithm 2.

---

**Algorithm 2** Parameter-Free Adam (Adam++)

---

1: **input:** $\mathbf{x}_0, \eta_0 = \epsilon, \delta, \beta_1, \beta_2, \lambda$
2: **for** $t = 0$, **to** $n$ **do**
3:     $r_t = \|\mathbf{x}_t - \mathbf{x}_0\|_2 / \sqrt{d}$
4:     $\eta_t = \max(\eta_{t-1}, r_t)$
5:     $\mathbf{g}_t = \mathcal{G}(\mathbf{x}_t)$
6:     $\beta_{1t} = \beta_1 \lambda^{t-1}$
7:     $\mathbf{m}_t = \beta_{1t}\mathbf{m}_{t-1} + (1 - \beta_{1t})\mathbf{g}_t$
8:     **Case 1:** $\mathbf{s}_t = (\sum_{i=0}^{t} \mathbf{g}_i^2)^{1/2}$
9:     **Case 2:** $\mathbf{v}_t = \beta_2 \mathbf{v}_{t-1} + (1 - \beta_2)\mathbf{g}_t^2, \mathbf{s}_t = \sqrt{(t+1) \cdot \max_{t' \leq t}(\mathbf{v}_{t'})}$
10:     $\mathbf{H}_t = \delta + \text{diag}(\mathbf{s}_t)$
11:     $\mathbf{x}_{t+1} = \mathbf{x}_t - \eta_t \cdot \mathbf{H}_t^{-1}\mathbf{m}_t$
12: **end for**

---

There are several key points in Algorithm 2 to note. First of all, Adam++ also implements the key quantity $r_t = \|\mathbf{x}_t - \mathbf{x}_0\|_2 / \sqrt{d}$ introduced in AdaGrad++ to automatically adapt the "learning rate". Moreover, Adam++ allows dynamically decaying first-moment parameter $\beta_{1t} = \beta_1 \lambda^t$, which follows the definition in AMSGrad (Reddi et al., 2018). When setting $\lambda = 1$, we can recover the common setup with a constant $\beta_1$. The introduction of the decaying $\beta_{1t}$ is due to technical reasons, and our theoretical analysis on Adam relies on a $\lambda \in (0, 1)$. However, we remark that Adam++ with $\lambda = 1$ can achieve highly competitive performance under various practical settings.

Another key feature of Adam++ is that it covers two cases. In **Case 1**, we implement entry-wise adaptive learning rates that are similar to AdaGrad and AdaGrad++. In **Case 2**, we implement a more common exponential moving average of the second moment $\mathbf{v}_t$ but also introduce another quantity $\mathbf{s}_t$. Particularly regarding the definition of $\mathbf{s}_t = \sqrt{(t+1) \cdot \max_{t' \leq t}(\mathbf{v}_{t'})}$, we note that the factor $\sqrt{(t+1)}$ ensures reasonable scaling when incorporated with the quantity $r_t$. This factor makes the scaling of $\mathbf{s}_t$ in **Case 2** more compatible with that in **Case 1**. Moreover, the max operation $\max_{t' \leq t}(\mathbf{v}_{t'})$ is inherited from the AMSGrad modification to Adam (Reddi et al., 2018), which has been shown to be crucial in ensuring theoretical guarantees. However, experiments have demonstrated that the simplified version $\mathbf{s}_t = \sqrt{(t+1) \cdot \mathbf{v}_t}$ works better in practice. This is consistent with many empirical observations (Gugger & Howard, 2018).

### 5.2 CONVERGENCE GUARANTEE OF ADAM++

In this section, we give the convergence guarantee of Adam++. The main result is given in the following theorem.

**Theorem 5.1.** Let $\mathbf{x}_0, \ldots, \mathbf{x}_T$ be the iterations of Adam++ following either **Case 1** or **Case 2** in Algorithm 2. In addition, let $\tau \in \arg\max_{t \leq T} \sum_{i=0}^{t-1} \frac{\eta_i}{\eta_t}$ and define $\overline{\mathbf{x}}_\tau = \frac{\sum_{t=0}^{T-1} \eta_t \mathbf{x}_\tau}{\sum_{t=0}^{T-1} \eta_t}$. Suppose $0 < \beta_1 < \sqrt{\beta_2}$ and $0 < \lambda < 1$. Then under Assumption 4.1, for any $\delta \in (0, 1)$, $L > 0$ and any $\mathbf{x}^* \in \mathbb{R}^d$, with probability at least $1 - \delta - \mathbb{P}(\max_{t \leq T} l(\mathbf{x}_t) > L)$, the following results hold:

$$f(\overline{\mathbf{x}}_\tau) \leq f(\mathbf{x}_*) + O\left(\left(\frac{D_\tau^2 \sqrt{d} \cdot \|\mathbf{s}_\tau\|_2}{\eta_0 T} + \frac{\overline{D}_\tau \sqrt{\theta_{\tau,\delta} \|\mathbf{s}_\tau\|_2^2 + L^2 \theta_{\tau,\delta}^2}}{T}\right) \log\left(\frac{\eta_T}{\eta_0}\right)\right).$$

Theorem 5.1 gives the convergence guarantee for Adam++. To the best of our knowledge, this is the first convergence guarantee of a parameter-free version of Adam. Clearly, the bound in Theorem 5.1

is of the same form as in Theorem 4.2 for AdaGrad++. Therefore, our comments on Theorem 4.2 also apply to Theorem 5.1: the bound holds with high probability when $l(\cdot)$ is bounded along the optimization trajectory $\mathbf{x}_0, \ldots, \mathbf{x}_T$. Moreover, similar to the bound for AdaGrad++, the quantity $\|\mathbf{s}_\tau\|_2$ is a key quantity: when $l(\mathbf{x})$ is bounded, the worst-case bound of $\|\mathbf{s}_\tau\|_2$ is $O(\sqrt{T})$, leading to a $\widetilde{O}(1/\sqrt{T})$ convergence rate. However, if $\|\mathbf{s}_\tau\|_2 = O(T^{1/2-\alpha})$ for some $\alpha \in (0, 1/2)$, we can expect a faster convergence rate .

Clearly, we can also establish the counterparts of Corollaries 4.3 and 4.4 for Adam++. However, to avoid repetitions, here we only give the corollary below as the counterpart of Corollary 4.4. The counterpart of Corollary 4.3 can be obtained by setting $\alpha = 0$.

**Corollary 5.2.** Suppose that the assumptions in Theorem 5.1 hold. Further assume that there exist $G > 0$ such that $l(\mathbf{x}) \leq G$ and $\|\mathbf{s}_\tau\|_2 \leq G \cdot T^{1/2-\alpha}$ for some $\alpha \in [0, 1/2)$. Then for any $\mathbf{x}^* \in \mathbb{R}^d$, with probability at least $1 - \delta$, it holds that

$$f(\overline{\mathbf{x}}_\tau) \leq f(\mathbf{x}_*) + \widetilde{O}\left( \frac{D_\tau^2 G \cdot \sqrt{d}}{T^{1/2+\alpha}} \right),$$

where $D_\tau = \max_{t \leq \tau} \|\mathbf{x}_t - \mathbf{x}_*\|_\infty$.

# 6 EXPERIMENTS

In this section, we evaluate the performance of Adam++ across image classification and large language model pretraining tasks to test its efficacy. For image classification problems, we train models on the CIFAR-10 dataset (Krizhevsky et al., 2009). To demonstrate Adam++'s versatility and stability across different network structures, we apply it to neural network architectures including VGG16 (Simonyan & Zisserman, 2014), ResNet-18, and ResNet-50 (He et al., 2016). We use Adam as the baseline, and also compare Adam++ against two state-of-the-art parameter-free algorithms: D-Adaptation (Defazio & Mishchenko, 2023) and Prodigy (Mishchenko & Defazio, 2023). For large language model pretraining tasks, we use a reproduced GPT-2 model with 125M and 355M parameters respectively on the OpenWebText dataset (Gokaslan & Cohen, 2019). Our training settings are based on those from NanoGPT and Sophia (Liu et al., 2023). We omit the experiments for AdaGrad++ as we found it consistently underperforms compared to Adam and Adam++, despite being better than AdaGrad.

## 6.1 IMAGE CLASSIFICATION

We aim to compare the optimization algorithms in a setting with minimal or no parameter tuning. On ResNet-18 and ResNet-50, we run the baseline Adam optimizer with a default learning rate of $1e^{-3}$ and a coupled weight decay of $5e^{-4}$. However, for VGG16, the same learning rate fails to converge, so we adjusted to a smaller learning rate of $1e^{-4}$ for Adam. For all parameter-free algorithms, including DAdapt Adam, Prodigy, and Adam++, although there is no learning rate choice required, we set a base learning rate factor that can be applied on top of the adaptive learning rate, as introduced in Ivgi et al. (2023); Mishchenko & Defazio (2023); Defazio & Mishchenko (2023). We set this base learning rate to $1.0$ across all parameter-free algorithms, while keeping all other parameters consistent with those of Adam, ensuring a fair comparison. For model architectures, we modify the output dimensions of ResNet and VGG networks to 10 to align with the number of output classes. We provide a detailed list of all training parameters in Appendix D. Variations in settings, especially in weight decay, may prevent Prodigy and DAdapt Adam from achieving optimal performance, with potential convergence issues on VGG16. In contrast, the results demonstrate that our algorithm remains robust across all benchmarks, even without any parameter tuning.

**Constant Learning Rate Schedule** Figure 1 illustrates the training loss and test accuracy curves against training epochs on the CIFAR-10 dataset for various network architectures and algorithms. The task is challenging due to the use of a fixed learning rate throughout all epochs. For the Adam++ algorithm, both Case 1 and Case 2 are implemented, with additional implementation details available in Appendix D. On ResNet-18 and ResNet-50, there is a noticeable performance gap between D-Adapt Adam, Prodigy, and Adam++ (Case 1) when compared to Adam. Conversely, Adam++ (Case 2) either matches or surpasses Adam's performance. On VGG16, while D-Adapt Adam and Prodigy fail to show improvement, Adam++ achieves test accuracies nearly identical to Adam. Furthermore, Figure 1 also reveals that although the test accuracies of Adam++ and Adam with a constant learning rate are similar, the training loss of Adam++ decreases faster.

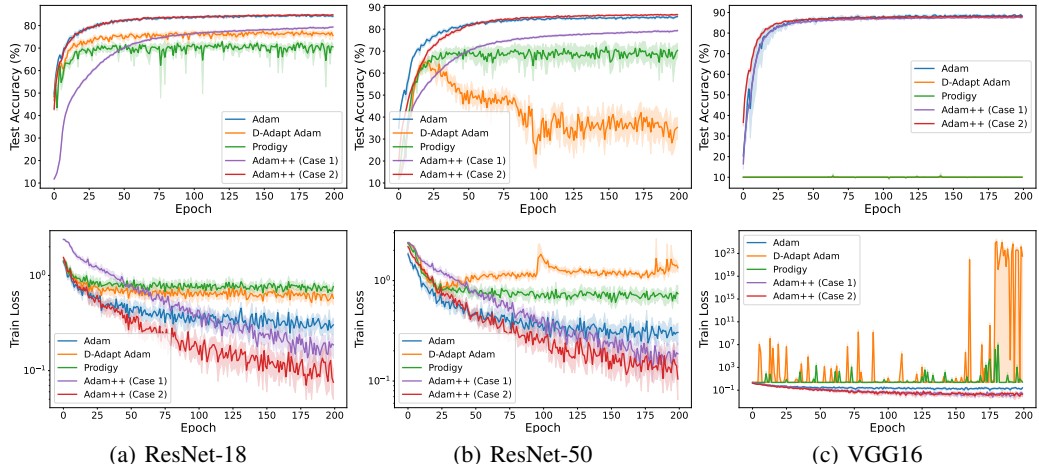

(a) ResNet-18      (b) ResNet-50      (c) VGG16

Figure 1: The results of training ResNet-18, ResNet-50, and VGG16 on CIFAR-10 with a constant learning rate schedule. Each curve represents the mean of 8 random runs, with the shaded area indicating the standard error. The first row presents the test accuracy of different algorithms, and the second row shows the training losses. Adam++ achieves performance superior or comparable to Adam.

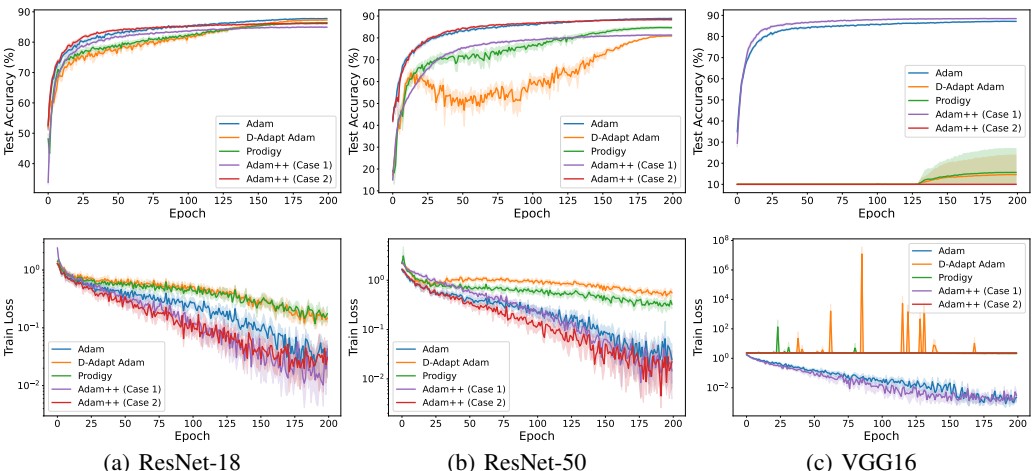

(a) ResNet-18      (b) ResNet-50      (c) VGG16

Figure 2: The results of training ResNet-18, ResNet-50, and VGG16 on CIFAR-10 with a cosine learning rate schedule. Each curve represents the mean of 8 random runs, with the shaded area indicating the standard error. The first row presents the test accuracy of different algorithms, and the second row shows the training losses.

**Cosine Learning Rate Schedule** In addition to the learning rates found by parameter-free algorithms, it is common to apply an additional learning rate schedule on top of that according to (Ivgi et al., 2023; Mishchenko & Defazio, 2023; Defazio & Mishchenko, 2023). Figure 2 provides a comparison of our algorithm with other baselines when utilizing the same cosine learning rate schedule. This annealed schedule aids in stabilizing training by being more cautious near the optimal point, thereby yielding better overall performance compared to a constant learning rate schedule. Under the annealed setting, both Prodigy and D-Adapt Adam exhibit improvement over their counterparts using a constant learning rate schedule. Notably, the performance enhancement becomes more pronounced in the later stages of training, suggesting that D-Adapt Adam and Prodigy might initially overestimate the learning rate. Meanwhile, our Adam++ algorithm maintains only a small gap with Adam. Notably, on VGG16, while the performance of D-Adapt Adam, Prodigy, and Adam++ (Case 2) fails to converge, Adam++ (Case 1) outperforms Adam.

We present the results for both constant and cosine learning rate schedules in Table 1. The reported values represent the best test accuracy or training loss achieved up to the final epoch. Notably, in nearly all cases, the top two algorithms in each row are either Adam or Adam++.

Table 1: Comparison of test accuracies and training losses for both constant and cosine Learning Rate Schedules on CIFAR-10 dataset.

| Model | LR schedule | Adam | | D-Adapt Adam | | Prodigy | | Adam++(case 1) | | Adam++(case 2) | |
|---|---|---|---|---|---|---|---|---|---|---|---|
| | | train loss | test acc | train loss | test acc | train loss | test acc | train loss | test acc | train loss | test acc |
| ResNet-18 | Constant | 0.0843 | **85.99** | 0.2696 | 80.13 | 0.4064 | 76.62 | **0.0360** | 80.5 | **0.0188** | 85.67 |
| | Cosine | **0.0015** | **88.03** | 0.0512 | 87.72 | 0.0416 | 86.83 | 0.0019 | 85.32 | 0.0018 | 86.6 |
| ResNet-50 | Constant | 0.0748 | 87.1 | 0.4983 | 70.3 | 0.3267 | 78.94 | 0.0340 | 81.2 | **0.0194** | **87.37** |
| | Cosine | 0.0017 | **89.1** | 0.2533 | 81.38 | 0.1226 | 86.02 | **0.0011** | 82.05 | 0.0013 | 88.74 |
| VGG16 | Constant | 0.0260 | **89.69** | 2.2638 | 10.56 | 2.2722 | 15.4 | **0.0004** | 88.36 | 0.0005 | 88.88 |
| | Cosine | **0.0001** | 87.47 | 1.2814 | 46.92 | 1.0421 | 55.23 | **0.0001** | 88.82 | 2.3016 | 10 |

## 6.2 LARGE LANGUAGE MODEL (LLM) PRETRAINING

In this subsection, we pretrain GPT-2 models with 125M and 355M parameters using the OpenWeb-Text dataset. For the baseline, we employ the AdamW optimizer instead of Adam, as empirically AdamW performs better than Adam in LLM tasks . For all parameter-free algorithms, including our proposed Adam++, we apply decoupled weight decay to align with AdamW, referring to the adjusted version of Adam++ as AdamW++. In detail, AdamW uses a standard cosine learning rate schedule with 2000 warm-up steps. The batch size is set to 480, with a learning rate of $6e^{-4}$ for GPT-2 small and $3e^{-4}$ for GPT-2 medium, as specified in Liu et al. (2023). All parameter-free algorithms use the same hyperparameters and learning rate schedule as AdamW. Additional details for pretraining are provided in Appendix D.

In Figures 3 and 4, we observe that AdamW++ outperforms AdamW by 0.02 in both training loss and validation loss on GPT-2 small and GPT-2 medium. In contrast, Prodigy performs 0.01 worse than AdamW on GPT-2 small and matches AdamW on GPT-2 medium, while D-Adapt Adam shows the weakest performance on these tasks. These results emphasize the ability of our algorithm to effectively handle large-scale language tasks.

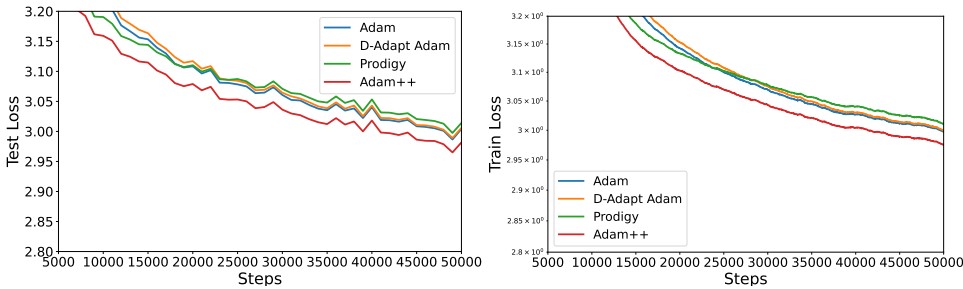

Figure 3: Comparison of training GPT-2 Small (155M) on OpenWebText. Left: Test loss. Performance at 50k steps—AdamW: 3.00, D-Adapt AdamW: 3.01, Prodigy: 3.01, Adam++: 2.98. Right: Train loss. Performance at 50k steps—AdamW: 2.97, D-Adapt AdamW: 2.97, Prodigy: 2.98, AdamW++: 2.95. AdamW++ refers to AdamW++ (Case 2).

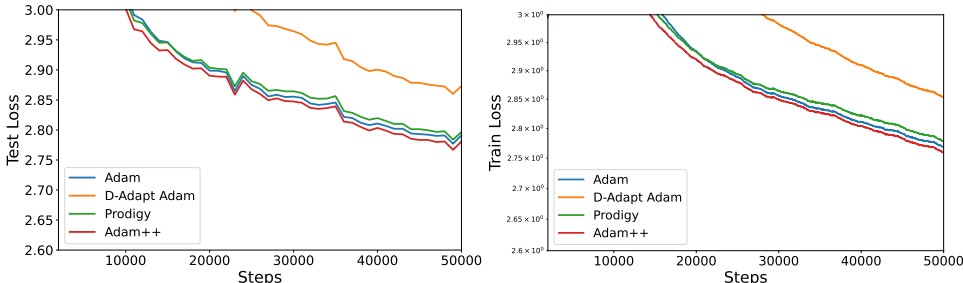

Figure 4: Comparison of training GPT-2 Medium (355M) on OpenWebText. Left: Test loss. Performance at 50k steps—AdamW: 2.80, D-Adapt AdamW: 2.87, Prodigy: 2.80, AdamW++: 2.78. Right: Train loss. Performance at 50k steps—AdamW: 2.75, D-Adapt AdamW: 2.82, Prodigy: 2.75, AdamW++: 2.73. AdamW++ refers to AdamW++ (Case 2).

### 6.3 ABLATION STUDY

We conduct an ablation study to assess the impact of different choices for the base learning rate and the initial learning rate on training loss and test accuracy using ResNet-50.

**Initial learning rate** $\eta_0$ Our theory suggests that the choice of the initial $\eta_0$ will not influence the final loss performance, as long as $\eta_0$ is not too large. We tested this hypothesis by running each of the problems using values of $\eta_0$ ranging from $10^{-6}$ to $1$. Figure 5 validates this conclusion in practice.

**Base learning rate** For this experiment alone, we consider Adam++ with different values of the base learning rate of $\eta_t = c \cdot \frac{\max_{i \leq t} \|\mathbf{x}_i - \mathbf{x}_0\|_2}{\sqrt{d}}$. According to our theory, our algorithms are expected to be unstable when $c > 1$ and slow to converge when $c < 1$. Figure 6 illustrates the performance around $c = 1$.

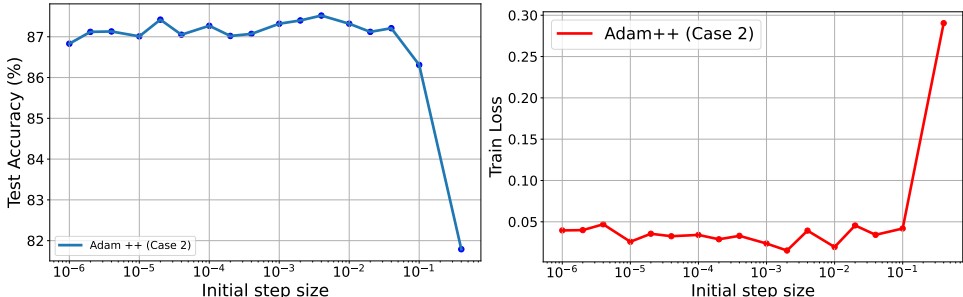

Figure 5: Effect of different choices of $\eta_0$ on test accuracy and training losses. When $\eta_0$ is less than $10^{-1}$, its influence on final performance is marginal.

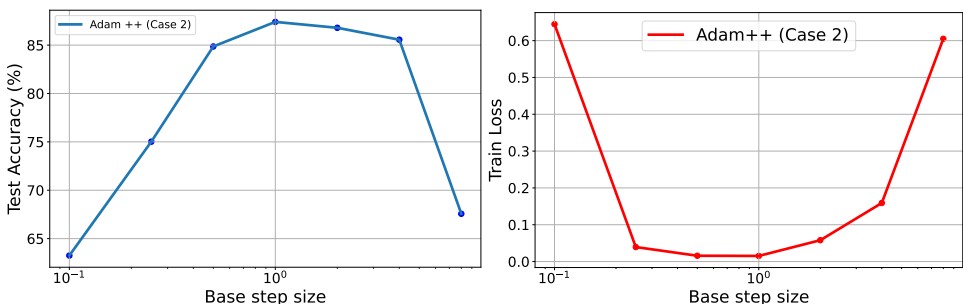

Figure 6: Effect of different choices of $c$ on test accuracy and training losses. When $c$ is between $0.5$ and $4$, its influence on final performance is limited.

## 7 CONCLUSIONS

In this paper, we propose two simple but effective algorithms, namely AdaGrad++ and Adam++, that are parameter-free variants of AdaGrad and Adam respectively. We demonstrate that, despite the simple intuition, AdaGrad++ and Adam++ are guaranteed to converge with a reasonable convergence rate, and also perform surprisingly well in various experiments. These theroetical and empirical results highlight the potential of AdaGrad++ and Adam++ to be robust and practical choices for a wide range of optimization tasks.

Several topics that are not covered in this paper are worth future studies. First of all, the current convergence analyses of AdaGrad++ and Adam++ are limited to the convex setting. Establishing convergence guarantees for AdaGrad++ and Adam++ under the setting of nonconvex optimization is an important future research direction. Moreover, establishing convergence guarantees for AdamW++ (which is used in our experiments without proof) is another promising area for future work.

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

## A    PROOF OF THEOREM 4.2

*Proof of Theorem 4.2.* We define $\mathbf{d}_t = \mathbf{x}_t - \mathbf{x}_*$, and define $\psi_t(\mathbf{x}) = \langle \mathbf{x}, \mathbf{H}_t\mathbf{x} \rangle$ and $B_\psi(\mathbf{x}, \mathbf{y}) = \psi(\mathbf{x} - \mathbf{y})/2$. Let $\mathbf{g}_t = (g_{t,1}, \cdots, g_{t,d}), \mathbf{s}_t = (s_{t,1}, \cdots, s_{t,d})$ and $\mathbf{d}_t = (d_{t,1}, \cdots, d_{t,d})$. From the definition of $\mathbf{x}_{t+1}$ we have

$$\mathbf{x}_{k+1} = \arg\min_{\mathbf{x}}\{\eta_k\langle\mathbf{g}_k, \mathbf{x}\rangle + B_{\psi_k}(\mathbf{x}, \mathbf{x}_k)\},$$

which gives

$$\langle\mathbf{x} - \mathbf{x}_{k+1}, \eta_k\mathbf{g}_k + \nabla\psi_k(\mathbf{x}_{k+1}) - \nabla\psi_k(\mathbf{x}_k)\rangle \geq 0 \tag{A.1}$$

for all $\mathbf{x}$. Setting $\mathbf{x} = \mathbf{x}^*$ and rearranging the terms, we can then obtain a bound of $\langle\mathbf{x}_{k+1} - \mathbf{x}^*, \mathbf{g}_k\rangle$. Thus we have the inequality by denoting the dual norm of $\|\cdot\|_{\psi_k}$ by $\|\cdot\|_{\psi_k^*}$

$$\eta_k\langle\mathbf{x}_k - \mathbf{x}^*, \mathbf{g}_k\rangle = \eta_k\langle\mathbf{x}_{k+1} - \mathbf{x}^*, \mathbf{g}_k\rangle + \eta_k\langle\mathbf{x}_k - \mathbf{x}_{k+1}, \mathbf{g}_k\rangle$$

$$\leq \langle\mathbf{x}^* - \mathbf{x}_{k+1}, \nabla\psi_k(\mathbf{x}_{k+1}) - \nabla\psi_k(\mathbf{x}_k)\rangle + B_{\psi_k}(\mathbf{x}_k, \mathbf{x}_{k+1}) + \frac{\eta_k^2}{2}\|\mathbf{g}_k\|_{\psi_k^*}^2$$

$$= B_{\psi_k}(\mathbf{x}^*, \mathbf{x}_k) - B_{\psi_k}(\mathbf{x}^*, \mathbf{x}_{k+1}) + \frac{\eta_k^2}{2}\|\mathbf{g}_k\|_{\psi_k^*}^2, \tag{A.2}$$

where the inequality follows by (A.1) and the Cauchy-Schwarz inequality for $\langle\mathbf{x}_k - \mathbf{x}_{k+1}, \eta_k\mathbf{g}_k\rangle$, and the second equality follows by the definition of $B_{\psi_k}(\mathbf{x}^*, \mathbf{x}_k), B_{\psi_k}(\mathbf{x}^*, \mathbf{x}_{k+1})$, and $B_{\psi_k}(\mathbf{x}_k, \mathbf{x}_{k+1})$. Further defining $\overline{\mathbf{x}}_t := \frac{1}{\sum_{k=0}^{t-1}\eta_k}\sum_{k=0}^{t-1}\eta_k\mathbf{x}_k$, and $\Delta_t := \nabla f(\mathbf{x}_t) - \mathbf{g}_t$, we have

$$f(\overline{\mathbf{x}}_t) - f(\mathbf{x}^*) \leq \frac{1}{\sum_{k=0}^{t-1}\eta_k}\sum_{k=0}^{t-1}\eta_k(f(\mathbf{x}_k) - f(\mathbf{x}^*))$$

$$\leq \frac{1}{\sum_{t=0}^{T-1}\eta_t}\sum_{t=0}^{T-1}\eta_t\big(\langle\mathbf{x}_t - \mathbf{x}_*, \mathbf{g}_t\rangle + \langle\mathbf{x}_t - \mathbf{x}_*, \nabla f(\mathbf{x}_t) - \mathbf{g}_t\rangle\big)$$

$$\leq \frac{1}{\sum_{t=0}^{T-1}\eta_t}\left\{\underbrace{\sum_{k=0}^{t-1}[B_{\psi_k}(\mathbf{x}^*, \mathbf{x}_k) - B_{\psi_k}(\mathbf{x}^*, \mathbf{x}_{k+1})]}_{I_1} + \underbrace{\frac{1}{2}\sum_{k=0}^{t-1}\eta_k^2\|\mathbf{g}_k\|_{\psi_k^*}^2}_{I_2}\right.$$

$$\left. + \underbrace{\sum_{k=0}^{t-1}\eta_k\langle\mathbf{x}_k - \mathbf{x}^*, \Delta_k\rangle}_{noise}\right\}, \tag{A.3}$$

where the first inequality follows by the convexity of $f(\mathbf{x})$ and Jensen's inequality, the second inequality follows again by the convexity of $f(\mathbf{x})$, and the last inequality follows by (A.2). For $I_1$ on the right-hand side of (A.3), we have

$$\sum_{k=0}^{t-1}B_{\psi_k}(\mathbf{x}^*, \mathbf{x}_k) - B_{\psi_k}(\mathbf{x}^*, \mathbf{x}_{k+1}) = \sum_{i=1}^{d}\sum_{k=0}^{t-1}s_{k,i}(d_{k,i}^2 - d_{k+1,i}^2)/2$$

$$\leq D_t^2\sum_{i=1}^{d}s_{t-1,i}. \tag{A.4}$$

Here, we use the fact of $D_t = \max_{i\leq t}\|\mathbf{x}_i - \mathbf{x}^*\|_\infty$,

For $I_2$ on the right-hand side of (A.3), we have

$$\sum_{k=0}^{t-1}\eta_k^2\|\mathbf{g}_t\|_{\psi_k^*}^2 \leq \eta_t^2\sum_{i=1}^{d}\sum_{k=0}^{t-1}\frac{g_{k,i}^2}{s_{k,i}} \leq 2\eta_t^2\sum_{i=1}^{d}s_{t-1,i} \leq O(D_t^2\sum_{i=1}^{d}s_{t-1,i}). \tag{A.5}$$

Here the first inequality holds for the nondecreasing of $\eta_t$, and the second inequality holds by using Lemma C.1 for every $i = 1, \cdots, d$. besides, noting that

$$\eta_t \leq \max_{k\leq t}\|\mathbf{x}_t - \mathbf{x}_0\|_2/\sqrt{d} + \epsilon \leq \max_{k\leq t}\|(\mathbf{x}_t - \mathbf{x}^*) - (\mathbf{x}_0 - \mathbf{x}^*)\|_2/\sqrt{d} + \epsilon \leq D_t \tag{A.6}$$

, thus we obtain the final inequality. For the *noise* of (A.3), let

$$Y_k = \eta_k \overline{D}_k, \ X_k = \left\langle \Delta_k, \frac{\mathbf{x}_k - \mathbf{x}_*}{\overline{D}_k} \right\rangle, \text{ and } \widehat{X}_k = -\left\langle \nabla f(\mathbf{x}_k), \frac{\mathbf{x}_k - \mathbf{x}_*}{\overline{D}_k} \right\rangle.$$

Thus we get

$$\sum_{k=0}^{t-1} Y_k X_k = \sum_{k=0}^{t-1} \eta_k \langle \Delta_k, \mathbf{x}_k - \mathbf{x}_* \rangle.$$

Therefore

$$\mathbb{P}\left( \exists t \leq T : \left| \sum_{k=0}^{t-1} \eta_k \langle \Delta_k, \mathbf{x}_k - \mathbf{x}_* \rangle \right| \geq 8\eta_{t-1}\overline{D}_{t-1} \sqrt{\theta_{t,\delta} \sum_{i=1}^{d} s_{t-1,i}^2 + L^2\theta_{t,\delta}^2} \right)$$

$$\leq \mathbb{P}\left( \exists t \leq T : \left| \sum_{k=0}^{t-1} Y_k X_k \right| \geq 8Y_t \sqrt{\theta_{t,\delta} \sum_{k=0}^{t-1} (X_{k-1} - \widehat{X}_{k-1})^2 + L^2\theta_{t,\delta}^2} \right) \leq \delta + \mathbb{P}(\bar{l}_T \geq L),$$

$$(A.7)$$

where the last inequality uses lemma C.2 and define $\bar{l}_T = \max_{t \leq T} l(\mathbf{x}_t)$.

By substituting (A.5),(A.4) and (A.7) into (A.3) we have that, for all $\delta \in (0,1)$ and $L > 0$, with probability at least $1 - \delta - \mathbb{P}(\bar{l}_T > L)$, for all $t \leq T$ the optimality gap $f(\overline{\mathbf{x}}_t) - f_*$ is

$$O\left( \frac{D_t^2 \sum_{i=1}^{d} s_{t,i}/\eta_t + 8\overline{D}_t \sqrt{\theta_{t,\delta} \sum_{i=1}^{d} s_{t,i}^2 + L^2\theta_{t,\delta}^2}}{\sum_{k=0}^{t-1} \eta_k/\eta_t} \right).$$

Further we use the QM-AM inequality to obtain the bound that $\|\mathbf{s}_t\|_1 \leq \sqrt{d}\|\mathbf{s}_t\|_2$. Finally, applying Lemma C.1 for $\frac{\eta_t}{\sum_{k=0}^{t-1} \eta_k}$ and using $\eta_0 < \eta_t$ bound $\eta_t$ on the molecule finishes the proof. $\quad\square$

## B  PROOF OF THEOREM 5.1

*Proof of Theorem 5.1.* We define $\mathbf{d}_t = \mathbf{x}_t - \mathbf{x}_*$, and let $\psi_t(\mathbf{x}) = \langle \mathbf{x}, \mathbf{H}_t \mathbf{x} \rangle$ and $B_\psi(\mathbf{x}, \mathbf{y}) = \psi(\mathbf{x} - \mathbf{y})/2$. Let $\mathbf{g}_t = (g_{t,1}, \cdots, g_{t,d}), \mathbf{s}_t = (s_{t,1}, \cdots, s_{t,d}), \mathbf{v}_t = (v_{t,1}, \cdots, v_{t,d})$ and $\mathbf{d}_t = (d_{t,1}, \cdots, d_{t,d})$. From the definition of $\mathbf{x}_{k+1}$ we have

$$\mathbf{x}_{k+1} = \arg\min_{\mathbf{x}}\{\eta_k \langle \mathbf{m}_k, \mathbf{x} \rangle + B_{\psi_k}(\mathbf{x}, \mathbf{x}_k)\},$$

which gives

$$\langle \mathbf{x} - \mathbf{x}_{k+1}, \eta_k \mathbf{m}_k + \nabla\psi_k(\mathbf{x}_{k+1}) - \nabla\psi_k(\mathbf{x}_k) \rangle \geq 0 \quad (B.1)$$

for all $\mathbf{x}$. Setting $\mathbf{x} = \mathbf{x}^*$ and rearranging the terms, we can then obtain a bound of $\langle \mathbf{x}_{k+1} - \mathbf{x}^*, \mathbf{m}_t \rangle$. Thus we have the inequality by denoting the dual norm of $\| \cdot \|_{\psi_k}$ by $\| \cdot \|_{\psi_k^*}$

$$\eta_k \langle \mathbf{x}_k - \mathbf{x}^*, \mathbf{m}_k \rangle = \eta_k \langle \mathbf{x}_{k+1} - \mathbf{x}^*, \mathbf{m}_k \rangle + \eta_k \langle \mathbf{x}_k - \mathbf{x}_{k+1}, \mathbf{m}_k \rangle$$

$$\leq \langle \mathbf{x}^* - \mathbf{x}_{k+1}, \nabla\psi_k(\mathbf{x}_{k+1}) - \nabla\psi_k(\mathbf{x}_k) \rangle + B_{\psi_k}(\mathbf{x}_k, \mathbf{x}_{k+1}) + \frac{\eta_k^2}{2}\|\mathbf{m}_k\|_{\psi_k^*}^2$$

$$= B_{\psi_k}(\mathbf{x}^*, \mathbf{x}_k) - B_{\psi_k}(\mathbf{x}^*, \mathbf{x}_{k+1}) + \frac{\eta_k^2}{2}\|\mathbf{m}_k\|_{\psi_k^*}^2,$$

where the inqeuality holds by (B.1) and the Cauchy-Schwarz inequality for $\langle \mathbf{x}_k - \mathbf{x}_{k+1}, \eta_k \mathbf{m}_k \rangle$. Using the fact that $\mathbf{m}_k = \beta_{1k}\mathbf{m}_{k-1} + (1 - \beta_{1k})\mathbf{g}_k$ we have

$$\eta_k \langle \mathbf{x}_k - \mathbf{x}^*, \mathbf{g}_k \rangle \leq \frac{1}{1 - \beta_{1k}}(B_{\psi_k}(\mathbf{x}^*, \mathbf{x}_k) - B_{\psi_k}(\mathbf{x}^*, \mathbf{x}_{k+1}))$$

$$+ \frac{\eta_k^2}{2(1 - \beta_1)}\|\mathbf{m}_k\|_{\psi_k^*}^2 - \frac{\eta_k \beta_{1k}}{1 - \beta_{1k}}\langle \mathbf{x}_k - \mathbf{x}^*, \mathbf{m}_{k-1} \rangle$$

$$\leq \frac{1}{1 - \beta_{1k}}(B_{\psi_k}(\mathbf{x}^*, \mathbf{x}_k) - B_{\psi_k}(\mathbf{x}^*, \mathbf{x}_{k+1}))$$

$$+ \frac{\eta_k^2}{2(1 - \beta_{1k})}\|\mathbf{m}_k\|_{\psi_k^*}^2 + \frac{\eta_k^2 \beta_{1k}}{2(1 - \beta_{1k})}\|\mathbf{m}_{k-1}\|_{\psi_k^*}^2 + \frac{\beta_{1k}}{1 - \beta_{1k}}B_{\psi_k}(\mathbf{x}_k, \mathbf{x}^*). \quad \text{(B.2)}$$

Noting that

$$f(\overline{\mathbf{x}}_t) - f(\mathbf{x}^*) \leq \frac{1}{\sum_{k=0}^{t-1} \eta_k} \sum_{k=0}^{t-1} \eta_k \langle \mathbf{x}_k - \mathbf{x}^*, \nabla f(\mathbf{x}_k) \rangle$$

$$= \frac{1}{\sum_{k=0}^{t-1} \eta_k}(\sum_{k=0}^{t-1} \eta_k \langle \mathbf{x}_k - \mathbf{x}^*, \mathbf{g}_k \rangle + \eta_k \langle \mathbf{x}_k - \mathbf{x}^*, \Delta_k \rangle),$$

where $\Delta_t = \nabla f(\mathbf{x}_t) - \mathbf{g}_t$, thus we can substitute (B.2) into it and lead to

$$f(\overline{\mathbf{x}}_t) - f(\mathbf{x}^*) \leq \frac{1}{\sum_{k=0}^{t-1} \eta_k}\left\{ \underbrace{\sum_{k=0}^{t-1} \frac{(B_{\psi_k}(\mathbf{x}^*, \mathbf{x}_k) - B_{\psi_k}(\mathbf{x}^*, \mathbf{x}_{k+1}))}{(1 - \beta_{1k})}}_{I_1} + \underbrace{\sum_{k=0}^{t-1} \frac{\beta_{1k}}{1 - \beta_{1k}}B_{\psi_k}(\mathbf{x}_k, \mathbf{x}^*)}_{I_2} \right.$$

$$+ \underbrace{\sum_{k=0}^{t-1}(\frac{\eta_k^2}{2(1 - \beta_{1k})}\|\mathbf{m}_k\|_{\psi_k^*}^2 + \frac{\eta_k^2 \beta_{1k}}{2(1 - \beta_{1k})}\|\mathbf{m}_{k-1}\|_{\psi_k^*}^2)}_{I_3}$$

$$\left. + \underbrace{\sum_{k=0}^{t-1} \eta_k \langle \mathbf{x}_k - \mathbf{x}^*, \Delta_k \rangle}_{\text{noise}} \right\}. \quad \text{(B.3)}$$

For $I_1$, we have

$$\sum_{k=0}^{t-1} \frac{B_{\psi_k}(\mathbf{x}^*, \mathbf{x}_k) - B_{\psi_k}(\mathbf{x}^*, \mathbf{x}_{k+1})}{1 - \beta_{1k}} \quad \text{(B.4)}$$

$$\leq \sum_{i=1}^{d} \sum_{k=0}^{t-1} \frac{s_{k,i}(d_{k,i}^2 - d_{k+1,i}^2)}{2(1 - \beta_1)}$$

$$= \sum_{i=1}^{d} \sum_{k=0}^{t-1} \frac{s_{k,i} D_t^2}{1 - \beta_1}. \quad \text{(B.5)}$$

Here the first inequality holds for the reason that $\beta_{1k} \leq \beta_1$, the second inequality holds for the definition of $D_t$ and thus $D_t > d_{k,i}$ for all $k < t$

For $I_2$, we have

$$B_{\psi_k}(\mathbf{x}_k, \mathbf{x}^*) \leq \frac{D_k^2}{2} \sum_{i=1}^{d} s_{k,i}.$$

And use the fact of $\beta_1 k = \beta_1 \lambda^k$ we have that

$$\sum_{k=0}^{t-1} \frac{\beta_{1k}}{1 - \beta_{1k}}B_{\psi_k}(\mathbf{x}_k, \mathbf{x}^*) \leq \frac{\beta_1 D_t^2}{2(1 - \beta_1)(1 - \lambda)} \sum_{i=1}^{d} s_{t-1,i}. \quad \text{(B.6)}$$

For $I_3$ in the inequality (B.3), we give the proofs for the two cases in Algorithm **3** separately.

**Case 1:** $\mathbf{s}_t = (\sum_{k=0}^{t} \mathbf{g}_k^2)^{1/2}$.
If we choose the first definition of $\mathbf{s}_t$ we have the fact that

$$\|\mathbf{m}_t\|_{\psi_t^*}^2 = \sum_{i=1}^{d} \frac{(\sum_{j=0}^{t}(1 - \beta_{1j})\Pi_{s=1}^{t-j}\beta_{1(t-s+1)}g_{j,i})^2}{\sqrt{\sum_{j=0}^{t} g_{j,i}^2}}$$

$$\leq \sum_{i=1}^{d} \frac{(\sum_{j=0}^{t} \Pi_{s=1}^{t-j} \beta_{1(t-s+1)})(\sum_{j=0}^{t} \Pi_{s=1}^{t-j} \beta_{1(t-s+1)} g_{j,i}^2)}{\sqrt{\sum_{j=0}^{t} g_{j,i}^2}}$$

$$\leq \sum_{i=1}^{d} \frac{(\sum_{j=0}^{t} \beta_1^{t-j})(\sum_{j=0}^{t} \beta_1^{t-j} g_{j,i}^2)}{\sqrt{\sum_{j=0}^{t} g_{j,i}^2}}$$

$$\leq \frac{1}{1-\beta_1} \sum_{i=1}^{d} \frac{\sum_{j=0}^{t} \beta_1^{t-j} g_{j,i}^2}{\sqrt{\sum_{j=0}^{t} g_{j,i}^2}}$$

The first inequality follows from Cauchy-Schwarz inequality. The second inequality is due to the fact that $\beta_{1j} \leq \beta_1$ for all $j \leq t$. The third inequality follows from the inequality $\sum_{j=1}^{t} \beta_1^{t-j} \leq 1/(1-\beta_1)$. Summarizing the inequalities we have

$$\sum_{k=0}^{t-1} \|\mathbf{m}_k\|_{\psi_k^*}^2 \leq \frac{1}{1-\beta_1} \sum_{i=1}^{d} \sum_{k=0}^{t-1} \frac{\sum_{j=1}^{k} \beta_1^{k-j} g_{j,i}^2}{\sqrt{\sum_{j=0}^{k} g_{j,i}^2}} = \frac{1}{1-\beta_1} \sum_{i=1}^{d} \sum_{k=0}^{t-1} \sum_{j=0}^{k} \frac{\beta_1^{k-j} g_{j,i}^2}{\sqrt{\sum_{s=0}^{k} g_{s,i}^2}}$$

$$\leq \frac{1}{1-\beta_1} \sum_{i=1}^{d} \sum_{k=0}^{t-1} \sum_{j=0}^{k} \frac{\beta_1^{k-j} g_{j,i}^2}{\sqrt{\sum_{s=0}^{j} g_{s,i}^2}} = \frac{1}{1-\beta_1} \sum_{i=1}^{d} \sum_{j=0}^{t-1} \sum_{k=j}^{t-1} \frac{\beta_1^{k-j} g_{j,i}^2}{\sqrt{\sum_{s=0}^{j} g_{s,i}^2}}.$$

Moreover, we have

$$\sum_{k=0}^{t-1} \frac{\sum_{j=1}^{k} \beta_1^{k-j} g_{j,i}^2}{\sqrt{\sum_{j=0}^{k} g_{j,i}^2}} = \sum_{k=0}^{t-1} \frac{\sum_{j=1}^{k} \beta_1^{k-j} g_{j,i}^2}{\sqrt{\sum_{s=0}^{k} g_{s,i}^2}} = \sum_{k=0}^{t-1} \sum_{j=1}^{k} \frac{\beta_1^{k-j} g_{j,i}^2}{\sqrt{\sum_{s=0}^{k} g_{s,i}^2}} \leq \sum_{k=0}^{t-1} \sum_{j=1}^{k} \frac{\beta_1^{k-j} g_{j,i}^2}{\sqrt{\sum_{s=0}^{j} g_{s,i}^2}}$$

$$= \sum_{k \geq j, 0 \leq k \leq t-1} \frac{\beta_1^{k-j} g_{j,i}^2}{\sqrt{\sum_{s=0}^{j} g_{s,i}^2}} = \sum_{j=0}^{t-1} \sum_{k=j}^{t-1} \frac{\beta_1^{k-j} g_{j,i}^2}{\sqrt{\sum_{s=0}^{j} g_{s,i}^2}} = \sum_{j=0}^{t-1} \sum_{k=0}^{t-1-j} \frac{\beta_1^{k} g_{j,i}^2}{\sqrt{\sum_{s=0}^{j} g_{s,i}^2}}$$

$$= \sum_{j=0}^{t-1} (\sum_{k=0}^{t-1-j} \beta_1^{k}) \frac{g_{j,i}^2}{\sqrt{\sum_{s=0}^{j} g_{s,i}^2}}.$$

Therefore, noting that $\frac{g_{j,i}^2}{\sqrt{\sum_{k=0}^{j} g_{j,i}^2}} \leq 2(\sqrt{\sum_{k=0}^{j} g_{j,i}^2} - \sqrt{\sum_{k=0}^{j-1} g_{j,i}^2})$, we have

$$\sum_{k=0}^{t-1} \|\mathbf{m}_k\|_{\psi_j^*}^2 \leq \frac{1}{1-\beta_1} \sum_{i=1}^{d} \sum_{j=0}^{t-1} (\sum_{k=0}^{t-1-j} \beta_1^{k}) \frac{g_{j,i}^2}{\sqrt{\sum_{s=0}^{j} g_{s,i}^2}}$$

$$\leq \frac{2}{1-\beta_1} \sum_{i=1}^{d} \sum_{j=0}^{t-1} \beta_1^{t-1-j} \sqrt{\sum_{s=0}^{j} g_{s,i}^2}.$$

$$\leq \frac{2}{(1-\beta_1)^2} \|\mathbf{s}_{t-1}\|_1 \tag{B.7}$$

**Case 2:** $\mathbf{s}_t = \sqrt{(t+1) \cdot \max_{k \leq t}(\mathbf{v}_k)}$.

If we choose the second form of $\mathbf{s}_t$, suppose $\gamma = \beta_1/\sqrt{\beta_2} < 1$, and we have

$$\|\mathbf{m}_k\|_{\psi_k^*}^2 = \sum_{i=1}^{d} \frac{m_{k,i}^2}{s_{k,i}} \leq \sum_{i=1}^{d} \frac{m_{k,i}^2}{\sqrt{(k+1) v_{k,i}}}$$

$$= \sum_{i=1}^{d} \frac{(\sum_{j=0}^{k} (1-\beta_{1j}) \Pi_{s=1}^{k-j} \beta_{1(k-s+1)} g_{j,i})^2}{\sqrt{(k+1)((1-\beta_2) \sum_{j=0}^{k} \beta_2^{k-j} g_{j,i}^2)}}$$

For the definition of $\mathbf{s}_t$. Further using the Cauchy-Schwarz inequality and the fact that $\beta_{1t} \le \beta_1$ we have

$$\|\mathbf{m}_k\|_{\psi_k^*}^2 \le \sum_{i=1}^d \frac{(\sum_{j=0}^k \Pi_{s=1}^{k-j} \beta_{1(k-s+1)})(\sum_{j=0}^k \Pi_{s=1}^{k-j} \beta_{1(k-s+1)} g_{j,i}^2)}{\sqrt{(k+1)((1-\beta_2)\sum_{j=0}^k \beta_2^{k-j} g_{j,i}^2}}$$

$$\le \sum_{i=1}^d \frac{(\sum_{j=0}^k \beta_1^{k-j})(\sum_{j=0}^k \beta_1^{k-j} g_{j,i}^2)}{\sqrt{(k+1)((1-\beta_2)\sum_{j=0}^k \beta_2^{k-j} g_{j,i}^2)}}.$$

And then for the inequality that $\sum_{j=0}^k \beta_1^{k-j} \le 1/(1-\beta_1)$ we have

$$\|\mathbf{m}_k\|_{\psi_k^*}^2 \le \frac{1}{(1-\beta_1)\sqrt{(k+1)(1-\beta_2)}} \sum_{i=1}^d \frac{\sum_{j=0}^k \beta_1^{k-j} g_{j,i}^2}{\sqrt{\sum_{j=0}^k \beta_2^{k-j} g_{j,i}^2}}$$

$$\le \frac{1}{(1-\beta_1)\sqrt{(k+1)(1-\beta_2)}} \sum_{i=1}^d \sum_{j=0}^t \frac{\beta_1^{k-j} g_{j,i}^2}{\sqrt{\beta_2^{k-j} g_{j,i}^2}}$$

$$\le \frac{1}{(1-\beta_1)\sqrt{(k+1)(1-\beta_2)}} \sum_{i=1}^d \sum_{j=0}^k \gamma^{k-j} |g_{j,i}|.$$

Thus the sum of $\|\mathbf{m}_t\|_{\psi_t^*}^2$ can further be bounded as follows:

$$\sum_{k=0}^t \|\mathbf{m}_k\|_{\psi_k^*}^2 \le \sum_{k=0}^t \frac{1}{(1-\beta_1)\sqrt{(k+1)(1-\beta_2)}} \sum_{i=1}^d \sum_{j=0}^k \gamma^{k-j} |g_{j,i}|$$

$$= \frac{1}{(1-\beta_1)\sqrt{1-\beta_2}} \sum_{i=1}^d \sum_{k=0}^t |g_{k,i}| \sum_{j=k}^t \frac{\gamma^{j-t}}{\sqrt{j+1}}$$

$$\le \frac{1}{(1-\beta_1)\sqrt{1-\beta_2}} \sum_{i=1}^d \sum_{k=0}^t |g_{k,i}| \sum_{j=k}^t \frac{\gamma^{j-k}}{\sqrt{k+1}}$$

$$\le \frac{1}{(1-\beta_1)\sqrt{1-\beta_2}} \sum_{k=0}^t \frac{\|\mathbf{g}_p\|_1}{(1-\gamma)\sqrt{(k+1)}} \tag{B.8}$$

For the noise term, we define $\overline{D}_t = \max k \le t \|\mathbf{d}_k\|_2$, and let

$$Y_k = \eta_k \overline{D}_k, \ X_k = \left\langle \Delta_k, \frac{\mathbf{x}_k - \mathbf{x}_*}{\overline{D}_k} \right\rangle, \text{ and } \widehat{X}_k = -\left\langle \nabla f(\mathbf{x}_k), \frac{\mathbf{x}_k - \mathbf{x}_*}{\overline{D}_k} \right\rangle.$$

Thus we get

$$\sum_{k=0}^{t-1} Y_k X_k = \sum_{k=0}^{t-1} \eta_k \langle \Delta_k, \mathbf{x}_k - \mathbf{x}_* \rangle.$$

Therefore

$$\mathbb{P}\left( \exists t \le T : \left| \sum_{k=0}^{t-1} \eta_k \langle \Delta_k, \mathbf{x}_k - \mathbf{x}_* \rangle \right| \ge 8\eta_{t-1} \overline{D}_{t-1} \sqrt{\theta_{t,\delta} \sum_{i=1}^d s_{t,i}^2 + L^2 \theta_{t,\delta}^2} \right)$$

$$\le \mathbb{P}\left( \exists t \le T : \left| \sum_{k=0}^{t-1} Y_k X_k \right| \ge 8Y_t \sqrt{\theta_{t,\delta} \sum_{k=0}^{t-1} (X_k - \widehat{X}_k)^2 + L^2 \theta_{t,\delta}^2} \right) \le \delta + \mathbb{P}(\bar{l}_T \ge L),$$

where the last inequality uses lemma C.2 and define $\bar{l}_t = \max_{k \leq t} l(\mathbf{x}_k)$. Therefore we have that, for all $\delta \in (0, 1)$ and $L > 0$, with probability at least $1 - \delta - \mathbb{P}(\bar{l}_T > L)$, for all $t \leq T$ we have

$$f(\overline{\mathbf{x}}_t) - f_* \leq (I_1 + I_2 + I_3) + \frac{8\overline{D}_t \eta_t}{\sum_{k=0}^{t-1} \eta_k} \sqrt{\theta_{t,\delta} \sum_{i=1}^{d} s_{t,i}^2 + L^2 \theta_{t,\delta}^2}. \qquad (B.9)$$

Thus substitute (B.5),(B.6) and(B.7)/(B.8) into (B.9) we have that for all $t < T$

$$f(\overline{\mathbf{x}}_t) - f_* \leq \frac{\eta_t}{\sum_{k=0}^{t-1} \eta_k} \left( \frac{D_t^2}{(1 - \beta_1)\eta_t} \|\mathbf{s}_{t-1}\|_1 + \frac{(1 + \beta_1)\eta_t}{(1 - \beta_1)^3} \|\mathbf{s}_{t-1}\|_1 \right.$$
$$\left. + \frac{\beta_1 D_t^2 \|\mathbf{s}_{t-1}\|_1}{2(1 - \beta_1)(1 - \lambda)\eta_t} + 8\overline{D}_t \sqrt{\theta_{t,\delta} \|\mathbf{s}_t\|_2^2 + L^2 \theta_{t,\delta}^2} \right)$$

for case 1, and

$$f(\overline{\mathbf{x}}_t) - f_* \leq \frac{\eta_t}{\sum_{k=0}^{t-1} \eta_k} \left( \frac{D_t^2}{(1 - \beta_1)\eta_t} \|\mathbf{s}_{t-1}\|_1 + \frac{(1 + \beta_1)\eta_t}{(1 - \beta_1)^2 \sqrt{1 - \beta_2}(1 - \gamma)} \sum_{k=0}^{t-1} \frac{\|\mathbf{g}_k\|_1}{\sqrt{k + 1}} \right.$$
$$\left. + \frac{\beta_1 D_t^2 \|\mathbf{s}_{t-1}\|_1}{2(1 - \beta_1)(1 - \lambda)\eta_t} + 8\overline{D}_t \sqrt{\theta_{t,\delta} \|\mathbf{s}_t\|_2^2 + L^2 \theta_{t,\delta}^2} \right)$$

for case 2 with probability at least $1 - \delta - \mathbb{P}(\bar{l}_T > L)$. We use the QM-AM inequality to obtain the bound that$\|\mathbf{s}_t\|_1 \leq \sqrt{d}\|\mathbf{s}_t\|_2$, and $\|\mathbf{g}_t\|_1 \leq \sqrt{d}\|\mathbf{g}_t\|_2$, and use (A.6) to bound $\eta_t$. Further we use Lemma C.1 for $\frac{\eta_t}{\sum_{k=0}^{t-1} \eta_k}$ and use $\eta_0 < \eta_t$ bound $\eta_t$ on the molecule and thus

$$f(\overline{\mathbf{x}}_\tau) - f^* \leq O\left( \log\left( \frac{\eta_T}{\eta_0} \right) \left( \frac{D_\tau^2 \sqrt{d}\|\mathbf{s}_\tau\|_2}{(1 - \beta_1)\eta_0} + \frac{(1 + \beta_1)D_\tau \sqrt{d}}{(1 - \beta_1)^3} \|\mathbf{s}_\tau\|_2 \right. \right.$$
$$\left. \left. + \frac{\beta_1 D_\tau^2 \sqrt{d}\|\mathbf{s}_\tau\|_2}{2(1 - \beta_1)(1 - \lambda)\eta_0} + 8\overline{D}_\tau \sqrt{\theta_{\tau,\delta} \|\mathbf{s}_\tau\|_2^2 + L^2 \theta_{\tau,\delta}^2} \right)/T \right)$$

for the Case 1 and

$$f(\overline{\mathbf{x}}_\tau) - f^* \leq O\left( \log\left( \frac{\eta_T}{\eta_0} \right) \left( \frac{D_\tau^2 \sqrt{d}\|\mathbf{s}_\tau\|_2}{(1 - \beta_1)\eta_0} + \frac{(1 + \beta_1)D_\tau \sqrt{d}}{(1 - \beta_1)^2 \sqrt{1 - \beta_2}(1 - \gamma)} \sum_{k=0}^{\tau-1} \frac{\|\mathbf{g}_k\|_2}{\sqrt{k + 1}} \right. \right.$$
$$\left. \left. + \frac{\beta_1 D_\tau^2 \sqrt{d}\|\mathbf{s}_\tau\|_2}{2(1 - \beta_1)(1 - \lambda)\eta_0} + 8\overline{D}_\tau \sqrt{\theta_{\tau,\delta} \|\mathbf{s}_\tau\|_2^2 + L^2 \theta_{\tau,\delta}^2} \right)/T \right)$$

for the Case 2 with probability at least $1 - \delta - \mathbb{P}(\bar{l}_T > L)$.

$\square$

## C  AUXILIARY LEMMAS

In this section, we present and summarize two auxiliary lemmas provided by Ivgi et al. (2023) that provide tools for our proof of the main theorems.

**Lemma C.1.** [Lemmas 3 and 4 in Ivgi et al. (2023)] Suppose that $0 < a_0 \leq a_1 \leq \cdots \leq a_T$. Then the following two inequalities hold:

$$\max_{t \leq T} \sum_{\tau < t} \frac{a_\tau}{a_t} \geq \frac{1}{e}\left( \frac{T}{\log_+(a_T/a_0)} - 1 \right), \qquad \sum_{k=1}^{t} \frac{a_k - a_{k-1}}{\sqrt{a_k}} \leq 2(\sqrt{a_t} - \sqrt{a_0}).$$

**Lemma C.2.** [Lemma 7 in Ivgi et al. (2023)] Consider a filtration $\mathcal{F} = \{\mathcal{F}_t\}_{t \geq 0}$ in a probability space. Let $S$ be the set of nonnegative and nondecreasing sequences. Suppose that $C_t \in \mathcal{F}_{t-1}$ and that $\{X_t\}_{t \geq 0}$ is a martingale difference sequence adapted to $\{\mathcal{F}_t\}_{t \geq 0}$ such that $|X_t| \leq C_t$ with

probability 1 for all $t \geq 0$. Then, for all $\delta \in (0,1)$, $c > 0$, $T > 0$, and $\overline{X}_t \in \mathcal{F}_{t-1}$ such that $|\overline{X}_t| \leq C_t$ with probability 1, it holds that

$$\mathbb{P}\left(\exists t \leq T, \exists \{y_i\}_{i=1}^{\infty} \in S \text{ such that } \left|\sum_{i=1}^{t} y_i X_i\right| \geq 8y_t \sqrt{\theta_{t,\delta} \sum_{i=1}^{t} (X_i - \overline{X}_i)^2 + c^2 \theta_{t,\delta}^2}\right)$$

$$\leq \delta + \mathbb{P}(\exists t \leq T : C_t \geq c),$$

where $\theta_{t,\delta} = \log(\frac{60 \log(6t)}{\delta})$.

# D    PARAMETER SETTINGS

Throughout the training, the base learning rate is fixed at $1.0$, and the initial learning rate of Adam++ is set to $1e^{-6}(1+\|x_0\|_2^2)$ for image classification tasks, as suggested by Ivgi et al. (2023). For GPT-2 small and GPT-2 medium tasks, the initial learning rates of AdamW++ are $6e^{-4}(1 + \|x_0\|_2^2)$ and $3e^{-4}(1 + \|x_0\|_2^2)$, respectively, where $6e^{-4}$ and $3e^{-4}$ correspond to the default learning rates for AdamW training. The initial learning rates of Prodigy and D-Adapt Adam are set as the default $1e^{-6}$ as the algorithms did not suggest any modification of this parameter.

In addition, we list the parameters, architectures and hardware that we used for the experiments. All other parameters not listed are set as default. The information is collected in Tables 2–3.

Table 2: CIFAR10 experiment.

| Hyper-parameter | Value |
|---|---|
| Architecture | ResNet 18, Resnet 50, VGG16 |
| Epochs | 200 |
| GPUs | 1×A100 |
| Batch size | 256 |
| LR schedule | Constant/Cosine Decay |
| Seeds | 1234+offset |
| weight decay | 5e-4 |
| Decoupled | No |
| $(\beta_1, \beta_2)$ | (0.9, 0.999) |
| Adam LR | 0.001 |

Table 3: Large language model experiment

| Hyper-parameter | Value |
|---|---|
| Architecture | GPT-2 Small/GPT-2 Medium |
| Steps | 50K |
| GPUs | 8×A100 |
| Batch size | 480 |
| Context Length | 1024 |
| LR schedule | Cosine Decay with Warmup |
| Seeds | 5000+offset |
| weight decay | 0.1 |
| Decoupled | yes |
| $(\beta_1, \beta_2)$ | (0.9, 0.95) |
| Adam LR | 6e-4/3e-4 |

# E    ADDITIONAL EXPERIMENTS

In this section, we present some additional experiment results.

## E.1    COMPARISONS TO ADAM WITH DIFFERENT INITIAL LEARNING RATES

Here, we present the results on training a VGG16 network on CIFAR-10 dataset with Adam implementing cosine learning rate schedule with different initial learning rates $\{1e-4, 5e-4, 1e-3\}$, and compare the results with the Adam++ algorihtm. All other hyperparameters are set according to the main paper, including a weight decay of $5e-4$.

Figure 7 shows the experiment results. These experiments are based on runs with 8 different random seeds, and both the mean and confidence intervals are shown in the plots. From the results, we can see that, as a parameter-free algorithm, Adam++ consistently delivers stable performance with $c = 1$ across various problems. Conversely, Adam requires meticulous tuning for each specific problem to achieve optimal results. For instance, when tuning Adam's learning rate for VGG16 within the range of $\{1e-4, 5e-4, 1e-3\}$, we found that Adam fails to converge at both $1e-3$ and $5e-4$. This result demonstrates that learning rate tuning is indeed important for Adam to achieve good performance and there is no simple golden choice of hyperparameters.

## E.2    RESULTS ON MORE NETWORK MODELS AND DATASETS

In this section, we expand our training to include additional network architectures and datasets. The architectures include a small Vision Transformer (ViT) with 820k parameters (Dosovitskiy, 2020), a

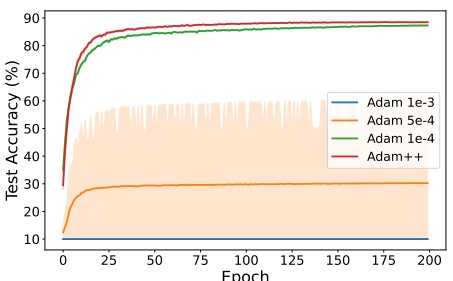 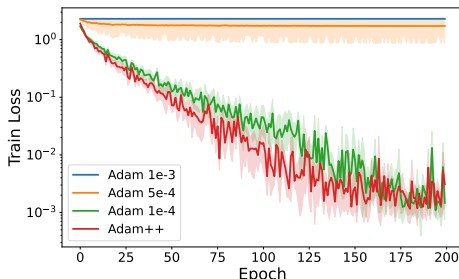

Figure 7: The results of training VGG16 on CIFAR-10 with cosine constant learning rate schedule. Each curve represents the mean of 8 random runs, with the shaded area indicating the standard error. The first figure shows the test accuracy, and the second figure shows the training losses. Based on these results, it is clear that Adam++ achieves performance superior or comparable to Adam with learning rate $1e-4$, but Adam with learning rate $1e-3$ or $5e-4$ fails to converge.

Wide ResNet-50-2 model (Zagoruyko, 2016), and a DenseNet-121 (Huang et al., 2017). Additionally, we train on the CIFAR-100 and SVHN datasets (Netzer et al., 2011). For hyperparameters, we run the baseline Adam optimizer with a default learning rate of $1e^{-3}$ with cosine decay schedule. For all parameter-free algorithms including D-Adapt Adam, Prodigy, and our Adam++, we set the base learning rate as 1.0 and use the same cosine decay schedule. All other hyperparameters remain consistent with those detailed in Section 6.1 and are unchanged throughout this section.

In Figure 8, 9 and 10, we present the performance comparisons across various models. For the Vision Transformer architecture, Adam++ (Case 1) consistently outperforms Adam. Notably, when training on the SVHN dataset, Adam, Prodigy, and D-Adapt Adam all fail to converge, whereas Adam++ (Case 1) achieves superior performance. In the case of the Wide ResNet-50-2, Adam++ (Case 2) surpasses the performance of Adam++ (Case 1), Prodigy, and D-Adapt Adam, and matches the performance of Adam with the default learning rate. For DenseNet-121, both Adam++ (Case 1) and Adam++ (Case 2) converge comparably to Adam. However, Prodigy and D-Adapt Adam exhibit weaker performance.

We note that throughout all image classification experiments, the base learning rate of Adam++ are kept as 1.0, while still demonstrating a strong and consistent performance across different scenarios. This consistency highlights Adam++'s robustness and its ability to perform reliably without the need for frequent adjustments or tuning.

### E.3 RESULTS ON ADAGRAD++

In Figures 11, 12, and 13, we evaluate the performance of AdaGrad++ in comparison to AdaGrad, Adam, Adam++, and other baselines including Prodigy and D-Adapt Adam. The hyperparameters for AdaGrad are aligned with those of Adam, using a learning rate of $1e^{-3}$ with a cosine decay schedule. While AdaGrad often significantly underperforms compared to Adam, AdaGrad++ demonstrates competitive or superior performance. It matches or exceeds the performance of both Adam and Adam++, and significantly outperforms AdaGrad.

### E.4 COMPUTATIONAL OVERHEAD

In this section, we analyze computational overhead by plotting training loss and test accuracy against wall-clock time, as shown in Figure 14. The results in Figure 14 are shown on the tasks of training ResNet-18 and ResNet-50 on CIFAR-10. Adam++ incurs less computational overhead compared to Prodigy and D-Adapt Adam, while delivering comparable or superior performance.

### E.5 ENLARGED VERSIONS OF FIGURES 1 AND 2

Here, we provide enlarged versions of Figures 1 and 2 to improve clarity. Figure 15 gives the enlarged figures in Figure 1, and Figure 16 gives the enlarged figures in Figure 2.

## F DISCUSSION ON THE MEMORY USAGE OF ADAGRAD++ AND ADAM++

In this section, we briefly discuss the memory usage of AdaGrad++ and Adam++. Specifically, we note that, compared to vanilla AdaGrad and Adam, AdaGrad++ and Adam++ require the storage

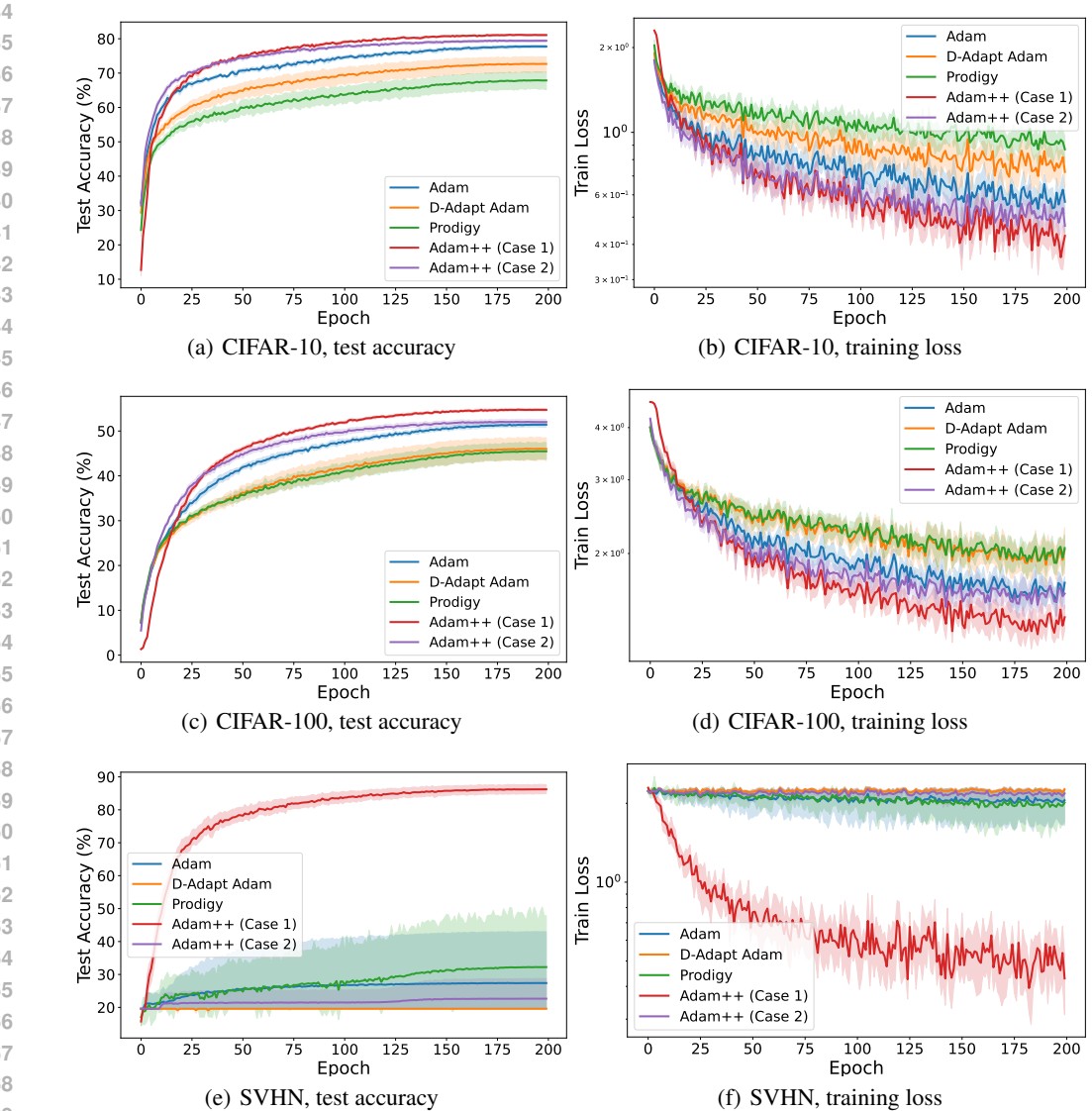

Figure 8: The results of training Vision Transformer on CIFAR-10, CIFAR-100, and SVHN with a consine learning rate schedule. Each curve represents the mean of 8 random runs, with the shaded area indicating the standard error. Adam++ achieves performance superior or comparable to Adam.

of an additional set of parameters, $\mathbf{x}_0$, resulting in slightly higher memory usage. However, it is important to highlight that, compared to existing parameter-free adaptive gradient methods such as Prodigy (Mishchenko & Defazio, 2023) and D-adaptation (Defazio & Mishchenko, 2023), which necessitate storing multiple intermediate quantities of the same size as the number of parameters, our proposed algorithms are more efficient in terms of memory usage.

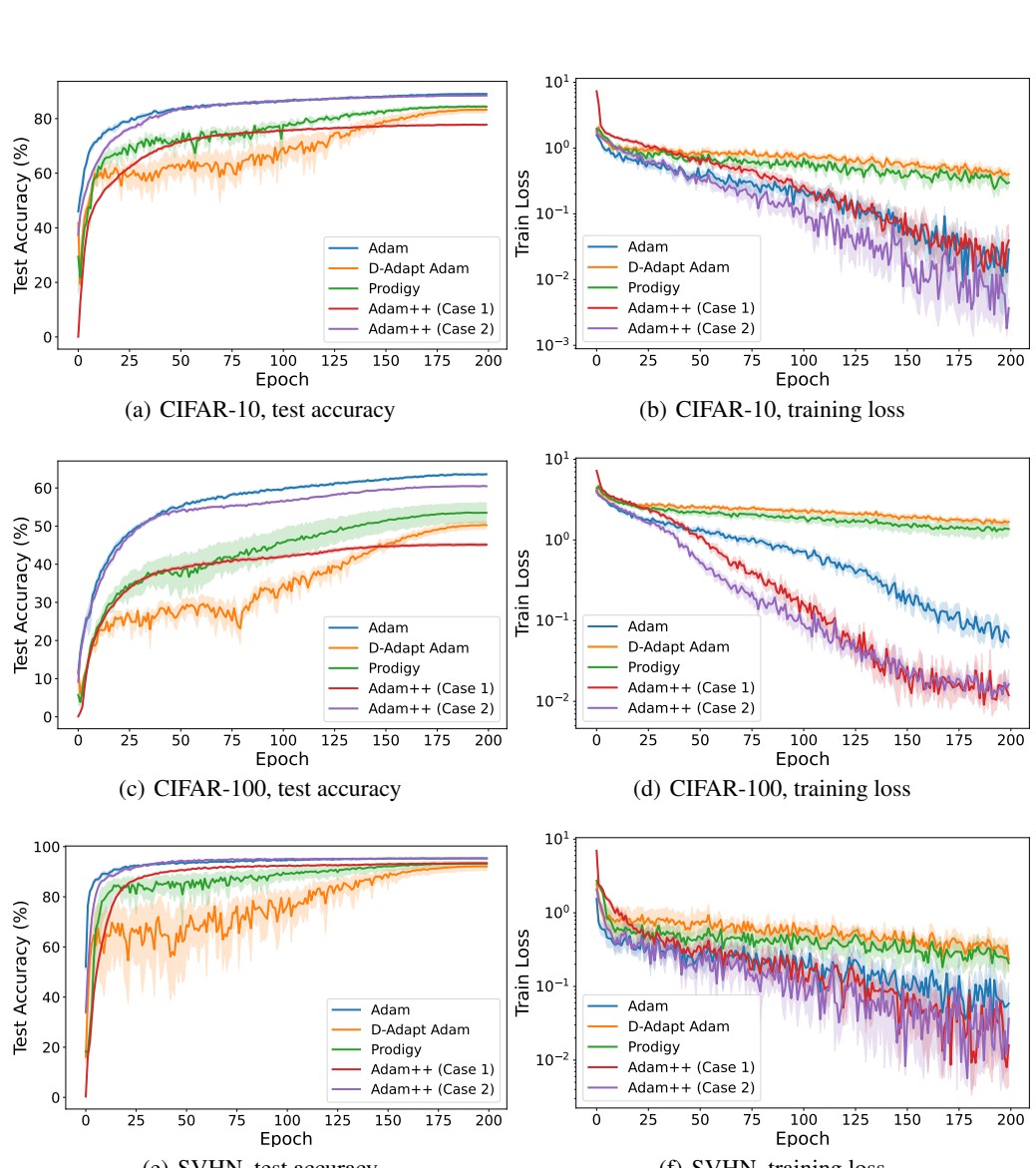

Figure 9: The results of training Wide ResNet-50-2 on CIFAR-10, CIFAR-100, and SVHN with a consine learning rate schedule. Each curve represents the mean of 8 random runs, with the shaded area indicating the standard error. Adam++ achieves performance superior or comparable to Adam.

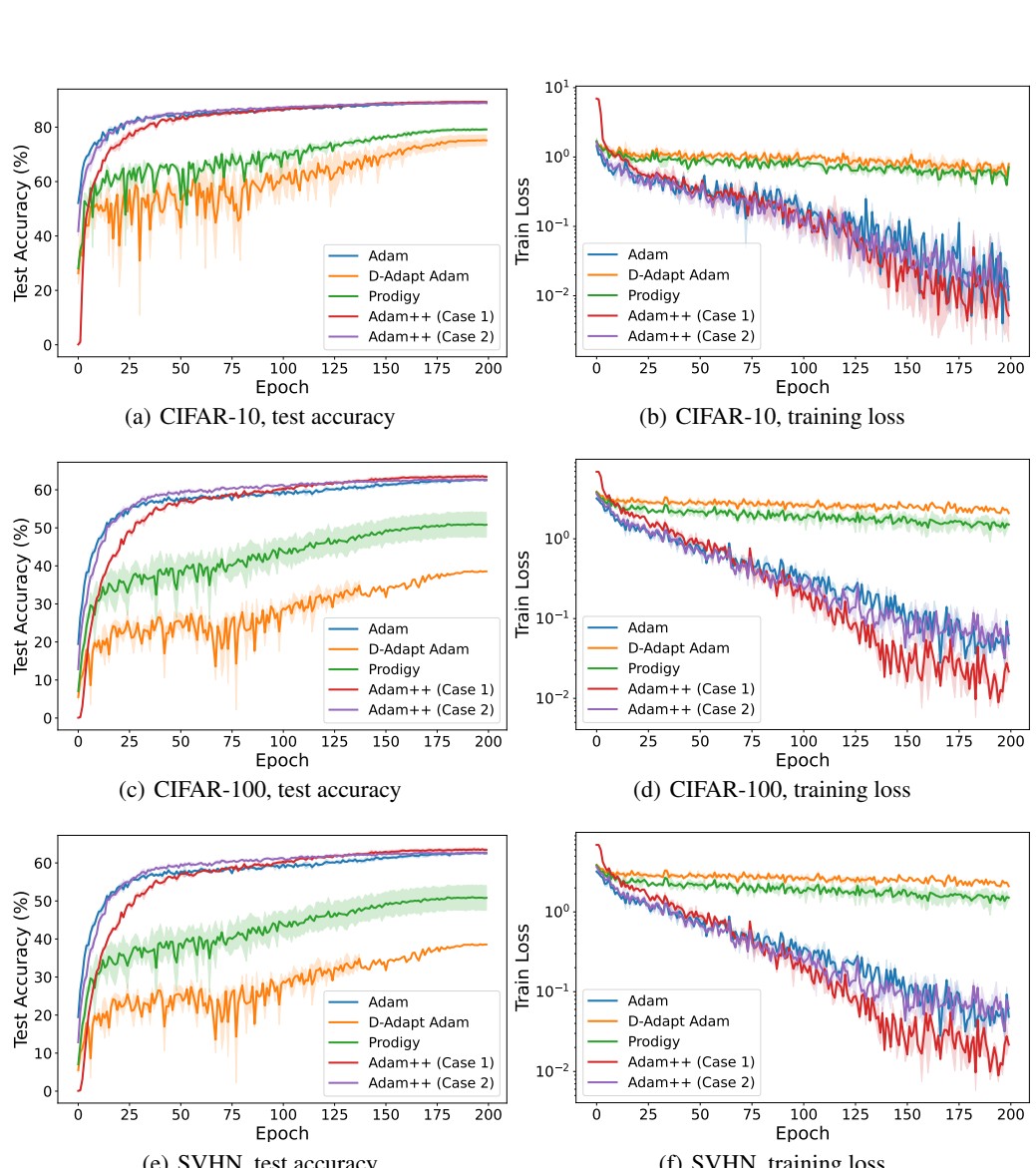

Figure 10: The results of training DenseNet-121 on CIFAR-10, CIFAR-100, and SVHN with a consine learning rate schedule. Each curve represents the mean of 8 random runs, with the shaded area indicating the standard error. Adam++ achieves performance superior or comparable to Adam.

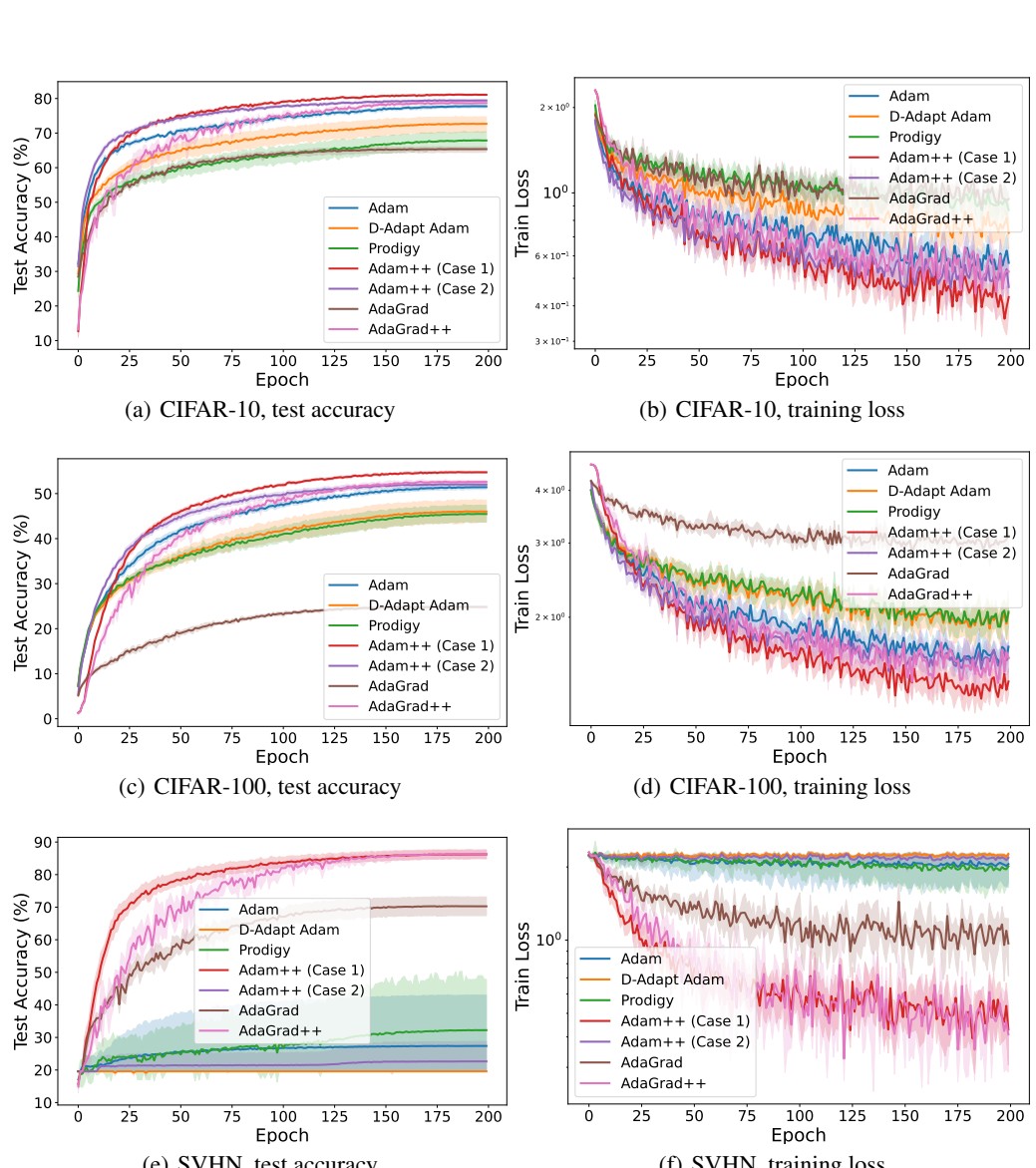

Figure 11: The performance comparison of various optimizers—Adam, D-Adapt Adam, Prodigy, Adam++ (Case 1), Adam++ (Case 2), AdaGrad, and AdaGrad++—conducted for training a Vision Transformer on CIFAR-10, CIFAR-100, and SVHN datasets using a cosine learning rate schedule.

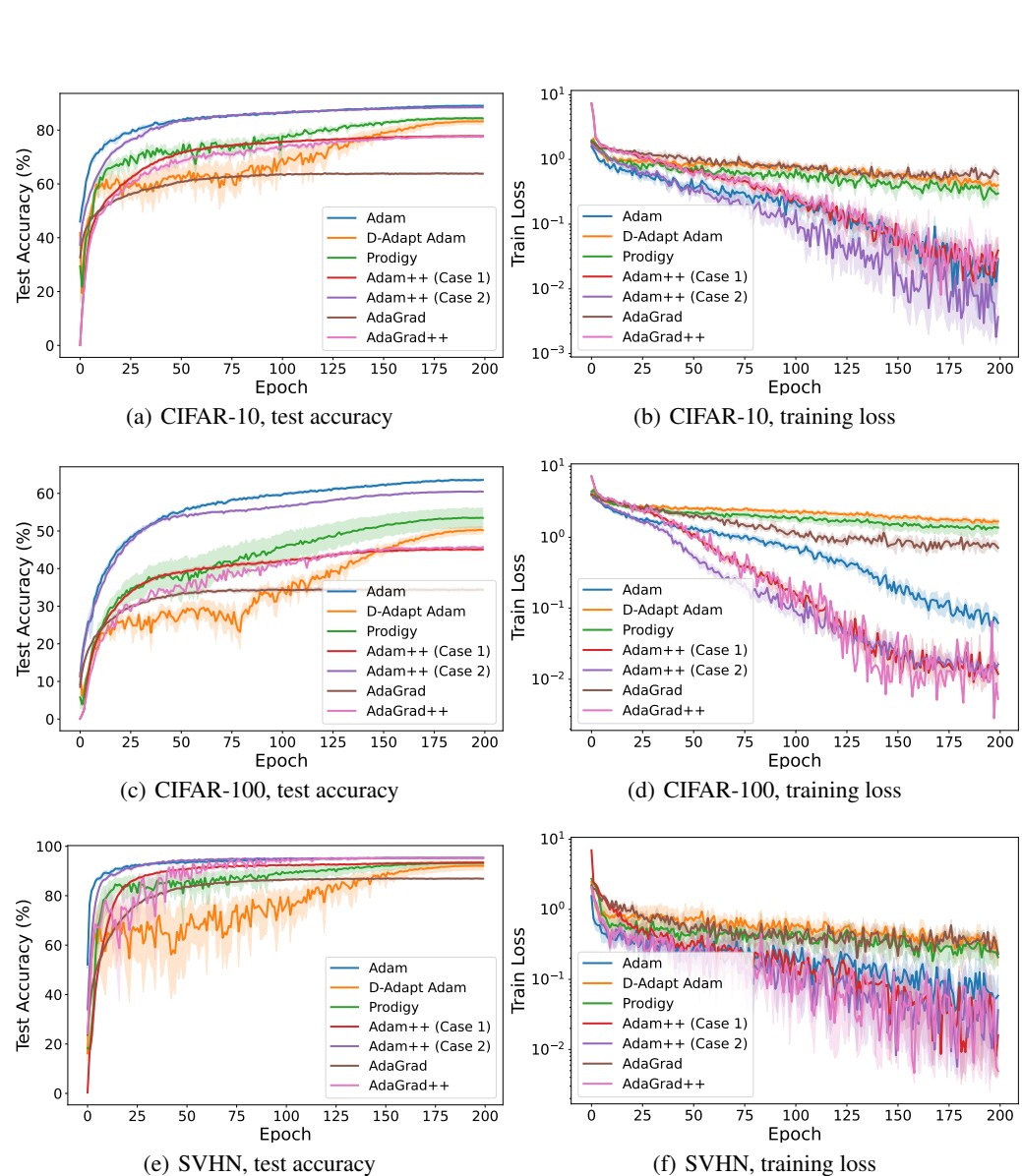

Figure 12: The performance comparison of various optimizers—Adam, D-Adapt Adam, Prodigy, Adam++ (Case 1), Adam++ (Case 2), AdaGrad, and AdaGrad++—conducted for training a Wide ResNet-50-2 on CIFAR-10, CIFAR-100, and SVHN datasets using a cosine learning rate schedule.

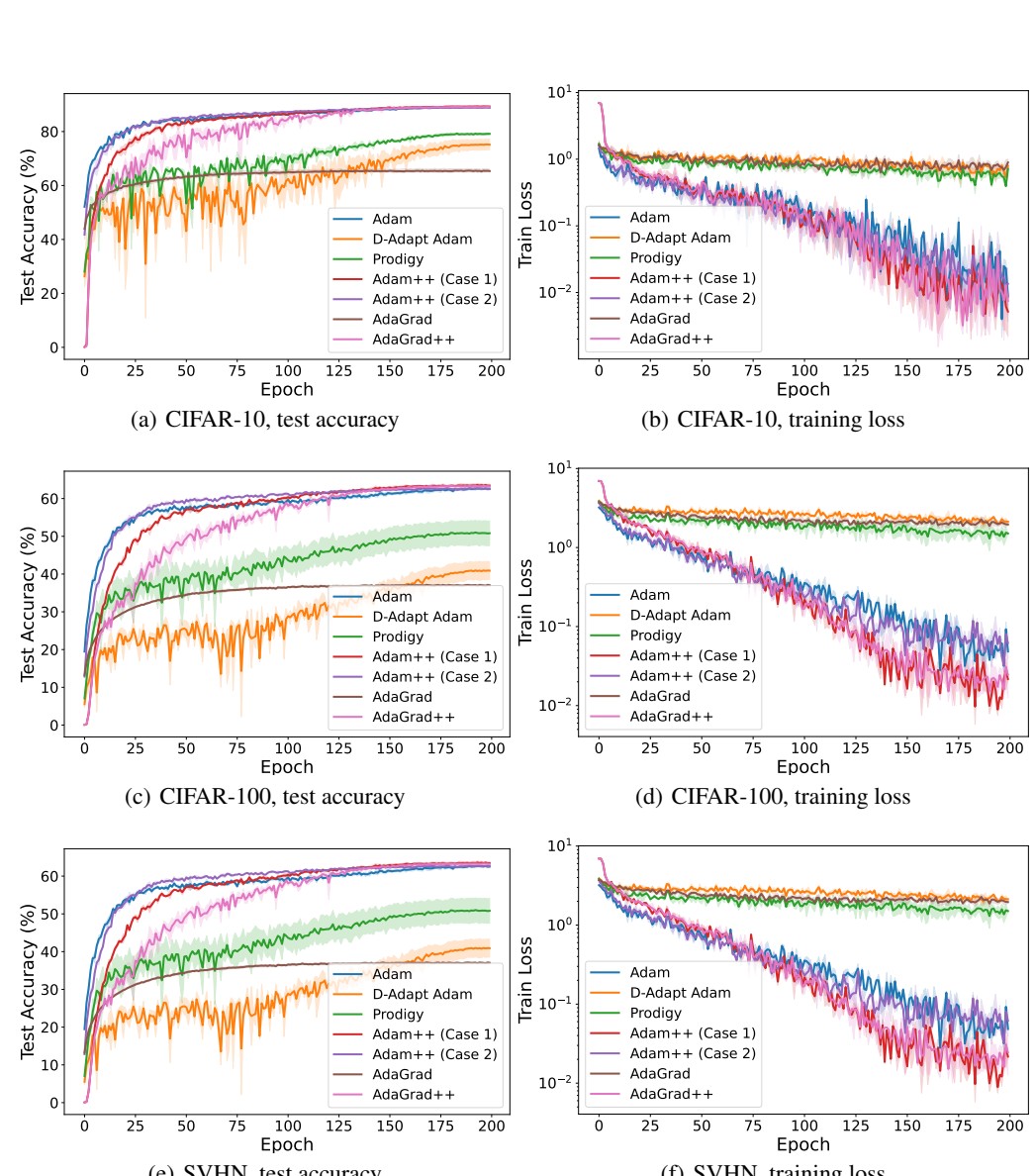

Figure 13: The performance comparison of various optimizers—Adam, D-Adapt Adam, Prodigy, Adam++ (Case 1), Adam++ (Case 2), AdaGrad, and AdaGrad++—conducted for training a DenseNet-121 on CIFAR-10, CIFAR-100, and SVHN datasets using a cosine learning rate schedule.

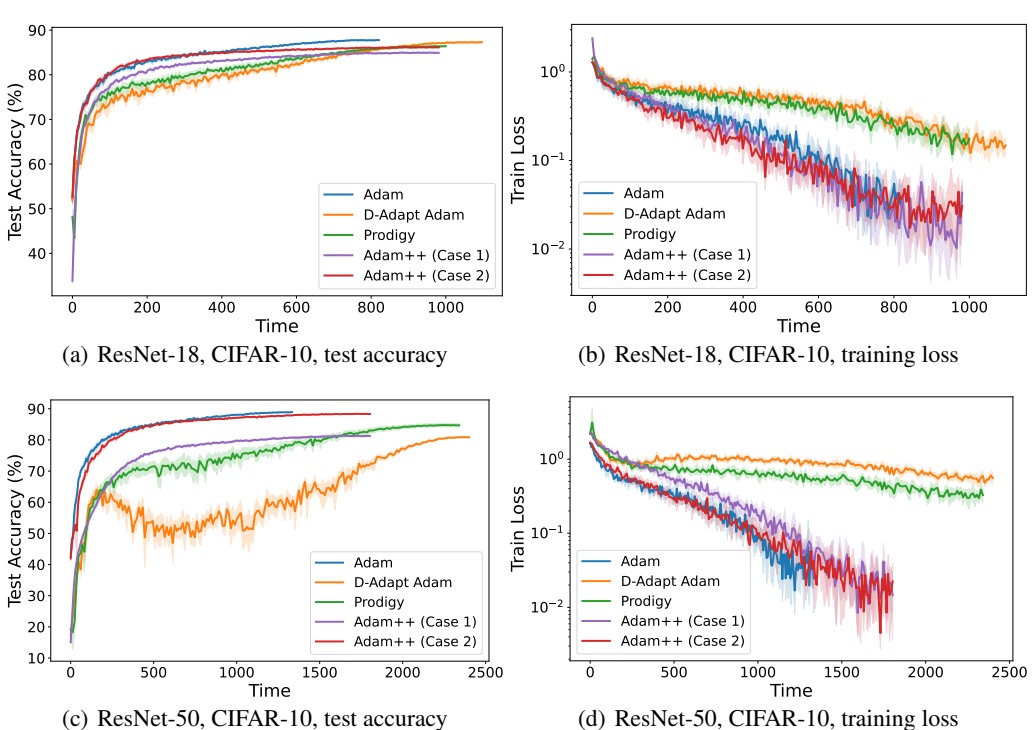

(a) ResNet-18, CIFAR-10, test accuracy

(b) ResNet-18, CIFAR-10, training loss

(c) ResNet-50, CIFAR-10, test accuracy

(d) ResNet-50, CIFAR-10, training loss

Figure 14: Test accuracy and training loss with respect to wall-clock time for different algorithms in training ResNet-18 and ResNet-50 on CIFAR-10 dataset.

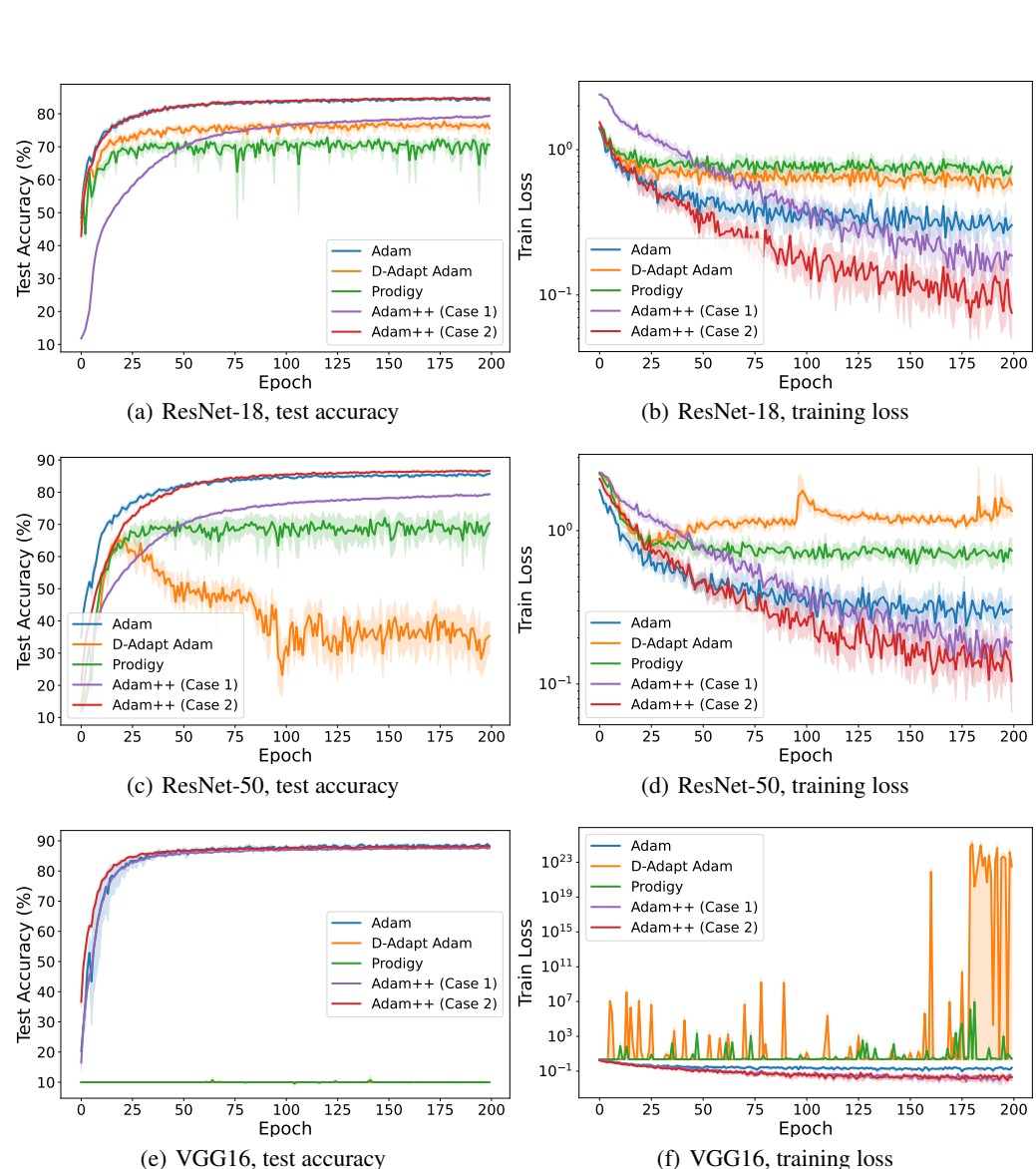

Figure 15: Enlarged version of Figure 1 on the results of training ResNet-18, ResNet-50, and VGG16 on CIFAR-10 with a constant learning rate schedule. Each curve represents the mean of 8 random runs, with the shaded area indicating the standard error. Adam++ achieves performance superior or comparable to Adam.

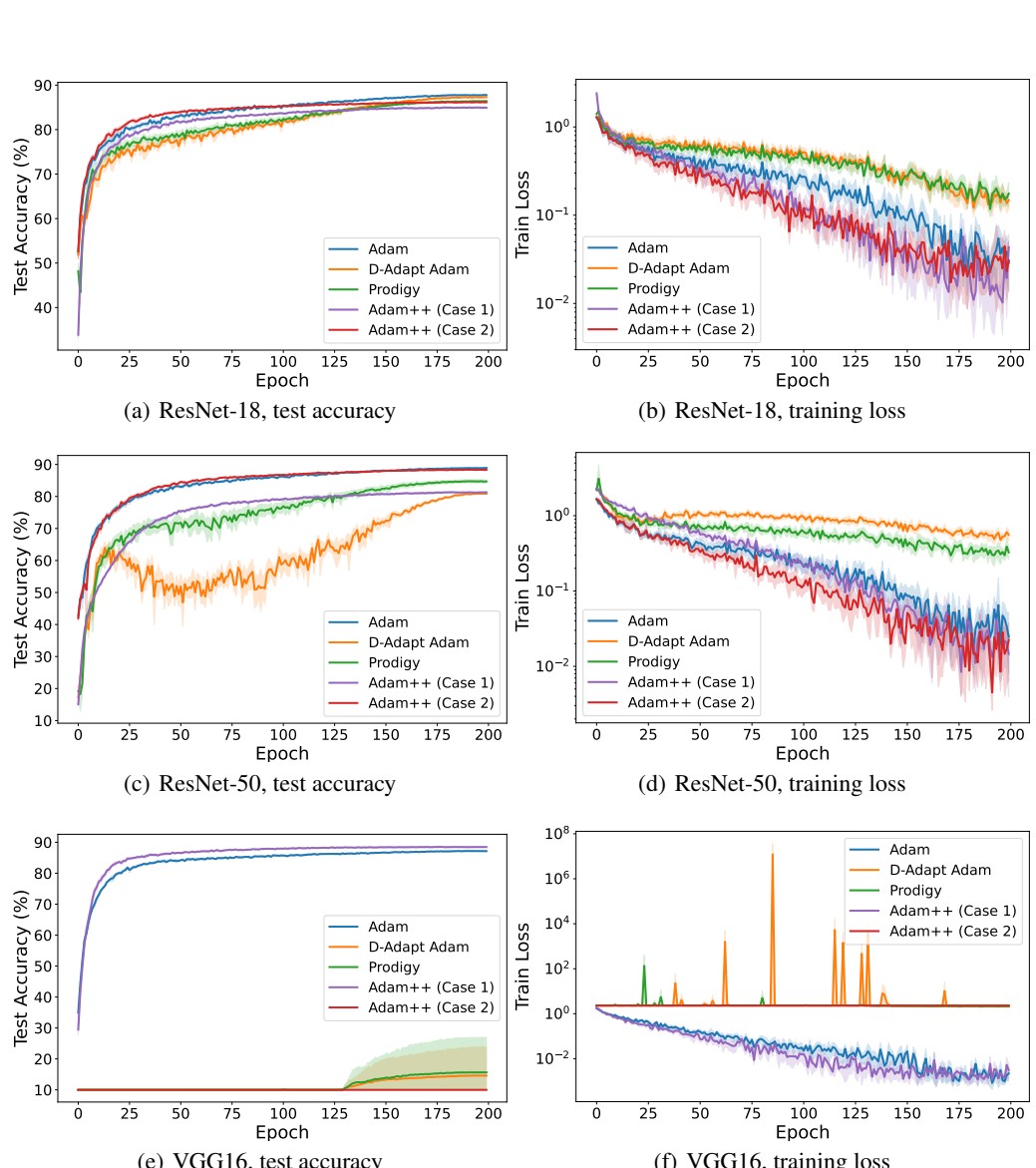

Figure 16: Enlarged version of Figure 2 on the results of training ResNet-18, ResNet-50, and VGG16 on CIFAR-10 with a cosine learning rate schedule. Each curve represents the mean of 8 random runs, with the shaded area indicating the standard error.

