# OpenReview forum: "Towards Simple and Provable Parameter-Free Adaptive Gradient Methods"
_ICLR.cc/2025/Conference — Submitted to ICLR 2025_

### Official Review · Reviewer_c1YW · 2024-10-15

**Soundness:** 3
**Presentation:** 3
**Contribution:** 2
**Rating:** 6
**Confidence:** 3

**Summary:**

The authors extend the idea of the DoG framework to Adam and AdaGrad to create parameter-free optimizers with adaptive learning rates. They also propose theorems for the convergence guarantees of their methods.

**Strengths:**

The paper is well-written, clear, and easy to follow. The authors extend the idea of the DoG framework to Adam and AdaGrad. They also propose theorems for the convergence guarantees of their methods; however, the math was not carefully checked by the reviewer.

**Weaknesses:**

-    The novel update rule of the learning rate requires storing in memory the original weights of the model, in addition to the weights of the current model and the first and second moments (in Adam’s case). This results in a memory overhead, approximately 33% larger than that of Adam. Furthermore, the update rules require the computation of an additional L2 norm, which can slowdown wallclock time for larger models. These memory and computational overheads are never mentioned or measured explicitly.

-    When comparing optimizers, wall-clock time is important; however, the paper does not mention how the proposed method compares with others in terms of computational time.

-    The minimal performance gains may not justify the added memory and computational overheads.

-    The plotted figures are so small that it is difficult to discern the details.

-    AdaGrad++ is introduced, but the authors state that it underperforms Adam and it is never actually plotted or shown in image classification or language modeling tasks. This raises the question of why it is included in the paper. A comparison with the standard AdaGrad version would have been appropriate since it was introduced.

-    Sophia is mentioned yet never compared against; the authors even state that they use NanoGPT from Sophia’s codebase, which is concerning. In fact, looking at Sophia’s results, it seems to the reviewer that it outperforms Adam++ on the GPT-2 language modeling tasks at 50K steps.

-    In image classification, standard Adam achieves the highest test accuracy in most cases.

-    CIFAR-10 is arguably an outdated task for image classification. It would be beneficial to test at least CIFAR-100, Tiny-ImageNet, and possibly full ImageNet. Additionally, testing transformer-based backbones like the ViT would be more appropriate than VGG16.

**Questions:**

-    What are the memory and computational overhead required for AdaGrad++ and Adam++?
-    How does Adam++ compare in wall-clock time with respect to classic Adam?
-    CIFAR-10 is a small dataset for today's standard, how does Adam++ perform on larger image classification datasets and on ViT backbones?

---

> ### Author Response · Authors · 2024-11-25
>
> We appreciate your constructive and helpful feedback. Regarding your questions and suggestions about additional experiments, we have followed your assumptions and added multiple new experiment results. We are currently working on some additional experiments, especially about comparisons between AdaGrad and AdaGrad++. We will update these results as soon as they are ready.
>
>
> >**Q1.** “The novel update rule of the learning rate requires storing in memory the original weights of the model, in addition to the weights of the current model and the first and second moments (in Adam’s case). This results in a memory overhead, approximately 33% larger than that of Adam. Furthermore, the update rules require the computation of an additional L2 norm, which can slowdown wallclock time for larger models. These memory and computational overheads are never mentioned or measured explicitly”, “What are the memory and computational overhead required for AdaGrad++ and Adam++?”
>
>
> **A1.** It is true that we need to store x_0 in memory. However, it is not true that memory overhead is approximately 33% larger than Adam.
> While AdamW stores $x_t$, $m_t$, $g_t$ and $v_t$, Adam++ includes an additional $x_0$, which accounts for an extra 25% memory usage. On the other hand, Prodigy and D-Adapt Adam both store $x_t$, $x_0$, $m_t$, $v_t$, $g_t$, $r_t$, $s_t$, $d_t$, taking up an extra 100% memory overheads over AdamW. This shows an advantage of Adam++ over previous parameter-free algorithms.
>
> Moreover, in tasks such as image classification, the memory demands of backpropagation typically exceed those for model storage. For instance, during the training of a DenseNet on CIFAR-10 (3*32*32) the memory usage statistics are as follows.
>
> |                | Adam++ | Adam | Prodigy | D-Adapt Adam |
> |----------------|--------|------|---------|--------------|
> | Memory Usage   | 1910Mb | 1874Mb | 1924Mb | 1900Mb       |
>
> The majority of the memory allocation is consumed by inputs, hidden layers, and the backpropagation process.
>
> Similar scenarios where model parameters are not the primary memory consumers also occur in training image diffusion models, graph NN/diffusion models, etc. In addition, LoRA fine-tuning also leads to low memory usage of model parameters. Under those circumstances, memory overhead of Adam++ are far less than 25% higher of AdamW.
>
>
>
> >**Q2.** “When comparing optimizers, wall-clock time is important; however, the paper does not mention how the proposed method compares with others in terms of computational time”, “How does Adam++ compare in wall-clock time with respect to classic Adam?”
>
> **A2.**
> Thank you for your suggestion. Adam++ shows a slightly longer computation time compared to Adam but remains competitive with other parameter-free algorithms. For instance, here is a comparison of the training times for 200 epochs on the CIFAR-10 dataset using a ResNet-18 model across different algorithms:
>
> |                  | Adam   | Adam++  | D-Adapt Adam | Prodigy |
> |------------------|--------|---------|--------------|---------|
> | Time (s)         | 825    | 986     | 1102         | 1005    |
>
> We will include a plot of the wall-clock time comparisons shortly in our revision.
>
>
>
>
>
>
> >**Q3.** The minimal performance gains may not justify the added memory and computational overheads.
>
> **A3.**  Please see our response in **A2.** for clarifications on memory and computational overheads. Our method has an advantage compared with other parameter-free algorithms.
>
> We would also like to clarify that one should not expect parameter-free methods to beat well-tuned Adam. The advantage of our algorithm (and other parameter-free algorithms) is not to beat Adam, but to minimize the tuning effort.
>
>
>
> >**Q4.** The plotted figures are so small that it is difficult to discern the details.
>
> **A4.** Thank you for your suggestion. Due to limited space in the main paper, we have provided enlarged versions of Figure 1 and Figure 2 in the appendix (please see Figure 15 and Figure 16 in the revised paper).
>
>
> >**Q5.** AdaGrad++ is introduced, but the authors state that it underperforms Adam and it is never actually plotted or shown in image classification or language modeling tasks. This raises the question of why it is included in the paper. A comparison with the standard AdaGrad version would have been appropriate since it was introduced.
>
> **A5.** Thank you for your suggestions! We are currently working on additional experiments about comparisons between AdaGrad and AdaGrad++. We will update these results as soon as they are ready.

---

> ### Author Response · Authors · 2024-11-25
>
> >**Q6.** Sophia is mentioned yet never compared against; the authors even state that they use NanoGPT from Sophia’s codebase, which is concerning. In fact, looking at Sophia’s results, it seems to the reviewer that it outperforms Adam++ on the GPT-2 language modeling tasks at 50K steps.
>
> **A6.** Our work focuses on developing efficient parameter-free adaptive gradient methods. It is true that we used Sophia’s codebase. As a second-order optimizer that is not parameter-free, Sophia is orthogonal to our contributions.
>
>
> >**Q7.** In image classification, standard Adam achieves the highest test accuracy in most cases.
>
> **A7.** As we have clarified in **A3**, the goal of parameter-free algorithms is not to beat well-tuned existing algorithms, but to reduce the effort required for tuning. As shown in Table 1, our proposed method outperforms parameter-free baselines convincingly.
>
>
> >**Q8.** “CIFAR-10 is arguably an outdated task for image classification. It would be beneficial to test at least CIFAR-100, Tiny-ImageNet, and possibly full ImageNet. Additionally, testing transformer-based backbones like the ViT would be more appropriate than VGG16”, “CIFAR-10 is a small dataset for today's standard, how does Adam++ perform on larger image classification datasets and on ViT backbones?
>
> **A8.** Thank you for your suggestions. We have expanded our revision to include two additional datasets: CIFAR-100 and SVHN, along with three additional model architectures: wide_resnet50_2, densenet121, and a small ViT model. We employed a learning rate of 1e-3 with a cosine decay for AdamW. For Prodigy, D-Adap Adam, and Adam++, we used a 1.0 base learning rate with cosine decay, maintaining these hyperparameters consistently across all experiments.
>
> As illustrated in Figure 8,9,10 in our revision, Adam++ demonstrates consistently stable performance across all benchmarks. In contrast, Prodigy and D-Adapt Adam show a need for some parameter tuning. Notably, there are scenarios where Prodigy, D-Adapt Adam, and Adam with a 1e-3 learning rate fail to perform effectively, such as with the ViT model on the SVHN dataset, whereas Adam++ maintains robust performance.
>
> We are still working on more experiment results and will update them as soon as they are available.

---

> ### Author Response · Authors · 2024-11-25
>
> Dear Reviewer,
>
> We have updated the paper again with new experimental results on (1) AdaGrad and AdaGrad++ (see Figures 11, 12 and 13), and (2) test accuracy and training loss curves with respect to wall-clock time (see Figure 14). As the discussion period is nearing its end, we sincerely hope you can review the extensive experimental results added in the revision. We believe these results address your concerns and demonstrate the performance of our proposed methods.
>
> If you have any further questions or comments, please let us know. Thanks.
>
> Best regards,
>
> Authors

---

> > ### Comment · Reviewer_c1YW · 2024-11-25
> >
> > Thank you for expanding the range of experiments, which partly addresses my concerns on the evaluation.
> > I have increased my score to 5, however I still have some fundamental concerns on the drawbacks of Adam++, like the increased memory requirements. Upon further inspection, prodigy does allow to tradeoff the memory overhead, and there exist other parameter free optimizers with same or fewer memory requirements than Adam, like ELRA [1]
> >
> > [1] Kleinsorge, Alexander, et al. "ELRA: Exponential learning rate adaption gradient descent optimization method." arXiv preprint arXiv:2309.06274 (2023).

---

> > > ### Author Response · Authors · 2024-11-26
> > >
> > > Dear Reviewer,
> > >
> > > We have cited and discussed the reference [1] you pointed out in the latest revision of the paper. However, it is important to note that [1] does not provide any theoretical convergence guarantees, and the experiments are conducted only on simple 2-dimensional optimization problems and 1-hidden layer neural networks on the MNIST dataset. In contrast, our work establishes rigorous theoretical guarantees for our proposed algorithms and demonstrates their practical value through extensive experiments, including training deep neural networks and GPT-2. Therefore, we believe it is unjustified to use [1] as a basis for diminishing the contributions of our work.
> > >
> > > Regarding your comment that prodigy allows for trading off memory overhead, we would like to point out prodigy-Adam actually requires more memory than our algorithms, as it needs to store the initial point ($\mathbf{x}_0$) along with additional intermediate variables ($\mathbf{r}_t, \mathbf{s}_t, \mathbf{d}_t$). In fact, to the best of our knowledge, there is no discussion of memory overhead in the Prodigy paper.
> > >
> > > After a thorough search, we discovered on their GitHub page that the authors introduced an engineering trick called slice_p to reduce memory usage by calculating LR adaptation statistics only on every p-th entry of each tensor. We assume this is what you're referring to. We would like to clarify that our proposed algorithms can certainly implement the same engineering trick to save memory.
> > >
> > > We would like to emphasize that the primary focus of this paper is to propose simple and efficient parameter-free algorithms that are backed by rigorous theoretical guarantees and competitive practical performance.
> > >
> > > Thank you!
> > >
> > > Best regards,
> > >
> > > Authors
> > >
> > >
> > > [1] Kleinsorge, Alexander, et al. "ELRA: Exponential learning rate adaption gradient descent optimization method." arXiv preprint arXiv:2309.06274 (2023).

---

> > > > ### Comment · Reviewer_c1YW · 2024-11-28
> > > >
> > > > Dear Authors
> > > >
> > > > I want to clarify that I'm not part of the paper I mentioned, so you are free to cite it or not, there are zero strings attached to it. Nonetheless, I appreciate the efforts you have put in addressing my concerns, I feel the paper is in a better shape now than it was at submission, so I'm increasing my score further to 6.

---

> > > > > ### Author Response · Authors · 2024-11-29
> > > > > **Thank you!**
> > > > >
> > > > > Thank you for raising your score. We're delighted that our rebuttal has addressed your questions and concerns. Your thoughtful comments and feedback have been invaluable in helping us improve the paper.

---

> ### Author Response · Authors · 2024-11-25
>
> Dear Reviewer,
>
> Thank you for your quick reply and for raising yoru score! Regarding your remaining concerns about the increased memory requirements, we would like to emphasize that we addressed this question in our original response **A1**. In particular, our experiments demonstrate that Adam++ does not really suffer from memory issues compared with other methods:
>
> |                | Adam++ | Adam | Prodigy | D-Adapt Adam |
> |----------------|--------|------|---------|--------------|
> | Memory Usage   | 1910Mb | 1874Mb | 1924Mb | 1900Mb       |
>
> Therefore, we are confident that our proposed algorithms have no particular disadvantage in terms of memory usage.
>
> We also appreciate the additional reference you pointed out, and we will cite it in our next revision. We would like to point out the additional reference [1] does not give theroetical guarantees, while we demonstrate that our proposed methods enjoy good theoretical convergence guarantees.
>
> Thanks! If you have any additional comments, please let us know.
>
> Best regards,
>
> Authors
>
>
> [1] Kleinsorge, Alexander, et al. "ELRA: Exponential learning rate adaption gradient descent optimization method." arXiv preprint arXiv:2309.06274 (2023).

---

### Official Review · Reviewer_C9og · 2024-10-17

**Soundness:** 2
**Presentation:** 2
**Contribution:** 1
**Rating:** 3
**Confidence:** 5

**Summary:**

This paper proposes a learning-rate-tuning-free method. Compared with the previous method DoG [1], this work introduces the parameter dimension $d$ into the Learning Rate (LR) and proposes AdaGrad++ and Adam++ by incorporating the new formula of LR into vanilla AdaGrad and Adam.

Besides, following the previous settings and proof framework of DoG, this work provides the convergence guarantee of the proposed methods, specifically, Theorem 4.2 and Theorem 5.2, under strong assumptions, i.e.,  convex, bounded stochastic gradient.  Regarding the evaluation, this work conducted image classification tasks - ResNet18, ResNet50, and VGG16 under CIFAR10 dataset, and language tasks - GPT-2 under OpenWebText dataset.

[1] Ivgi, Maor, Oliver Hinder, and Yair Carmon. "Dog is sgd’s best friend: A parameter-free dynamic step size schedule." International Conference on Machine Learning. PMLR, 2023.

**Strengths:**

The parameter-free methods have gained attention recently.  Achieving optimal convergence guarantees without the knowledge of specific problem-dependent properties is beneficial theoretically and practically.  And this paper tries to tackle this challenge practically and shows some evidence.

**Weaknesses:**

$\textbf{Weakness in techniques:}$

(1) Both Corollary 4.4 and Corollary 5.2 (and Theorem 4.2 and Theorem 5.1) state that the number of iterations requires $T \propto \mathcal{O}(d)$ to converge, where $d$ is the parameter dimension. Thus, this result is less useful or enlightening to demonstrate the practical performance of the proposed method since the $T<<d$ in practical practice. Besides, $T$ is independent of $d$ in previous work such as Corollary 1 of [1] and Theorem 1 of [2].

(2) Corollary 4.4 and Corollary 5.2 state similar convergence rates for AdaGrad++ and Adam++. However, as mentioned by the author “AdaGrad++ as we found it consistently underperforms compared to Adam and Adam++”.

(3) The first discrepancy between theorem proof and practical algorithm. Theorem 5.1 (Corollary 5.2) requires $\lambda \in (0,1)$, which actually disables the momentum update of Adam very quickly, seeing line-6 and line-7. However, I believe the success of Adam and the proposed method in fact counts on the momentum update, as mentioned by the Author that “Adam++ with $\lambda$ = 1 can achieve highly competitive performance under various practical settings”.

(4) The second discrepancy between theorem proof and practical algorithm. “Case 2” operation in Algorithm 2 requires the max operation over $v_{t}$ to comfort the convergence proof, however, the practical algorithm needs to eliminate it, as mentioned by the author “experiments have demonstrated that the simplified version $s_{t} = \sqrt{ (t + 1)  v_{t}}$ works better in practice”.

Overall, the technical proofs and results are less connected with the practical performance.

$\textbf{Weakness in evaluation:}$

(1) Considering a parameter-free (or learning-rate-free) optimizer, it is expected to perform well on a spectrum of optimization tasks across different datasets, network architectures, and training parameters such as batch size, number of epochs, etc.  I recognize that it is hard to comprehensively verify the method and lacks a uniform criterion of comparison. But universal adaptivity matters for LR-tuning-free methods.

However, the conducted experiments in this work are insufficient to demonstrate the adaptivity compared with previous work. For example, in terms of the image classification tasks, Table 5 of [1] employed 5 diverse network architectures and 12 different datasets, while this work employed 2 diverse network architectures (ResNet* and VGG) and 1 dataset. Same for the language tasks, seeing Table 4 of [1]. The small-scale experiment cannot support the adaptivity of the proposed methods.

(2) I believe a golden choice of training settings is: Adam+LR(1e-3)+Cosine LR decay, which could be used as the baseline instead of a constant learning rate, since decreasing LR w.r.t. Iterations is well supported by many theoretical results. Table 1 in this paper demonstrates that the default choice of training method, Adam+LR(1e-3)+Cosine LR decay, outperforms the proposed methods.


$\textbf{Weakness in method novelty or enlightenment:}$

(1) The main modification compared with the previous method is introducing $\sqrt{d}$ into the learning rate, however, this modification is not well supported intuitively or theoretically.


[1] Ivgi, Maor, Oliver Hinder, and Yair Carmon. "Dog is sgd’s best friend: A parameter-free dynamic step size schedule." International Conference on Machine Learning. PMLR, 2023.

[2] Mishchenko, Konstantin, and Aaron Defazio. "Prodigy: An expeditiously adaptive parameter-free learner." arXiv preprint arXiv:2306.06101 (2023).

**Questions:**

(1) how to explain $T\propto d$? and how to connect the theoretical results with the performance improvements?

(2) Regarding the image classification task, Adam++ is employed, however, AdamW++ (Case 2) is employed for language tasks. Do you have any particular reasons for the switching&tuning operations?  I also noticed two missing settings, i.e., AdamW++ (Case 1) and constant learning rate, in language tasks.

---

> ### Author Response · Authors · 2024-11-21
>
> Thank you for your detailed and constructive feedback! We address your comments and questions as follows. Please note that due to character limits in each response, we have split our replies into several parts.
>
> >**Q1.** “Both Corollary 4.4 and Corollary 5.2 (and Theorem 4.2 and Theorem 5.1) state that the number of iterations requires $T \propto O(d)$ to converge, where $d$ is the parameter dimension. Thus, this result is less useful or enlightening to demonstrate the practical performance of the proposed method since the $T \ll d$ in practical practice. Besides, $T$ is independent of $d$ in previous work such as Corollary 1 of [1] and Theorem 1 of [2]”,  “How to explain $T \propto d$? and how to connect the theoretical results with the performance improvements?”
>
> **A1.** We believe this is a misunderstanding. Please note that our work establishes theoretical convergence guarantees for our proposed parameter-free adaptive gradient methods. To our knowledge, [1] and [2] only established theoretical guarantees for parameter-free variants of (stochastic) gradient descent. For example, Corollary 1 of [1] mentioned in your comment is exactly for SGD with the proposed DoG step size schedule. Theorem 1 of [2] mentioned in your comment is for Algorithm 1 in [1], which is “Prodigy (GD version)”.
>
> The dependence in d is common in adaptive gradient methods. For example, for convex objectives, the original AMSGrad paper [3] provides convergence bounds similar to our result, which require $T \propto O(d)$ to converge (see Theorem 4 and Corollary 1 in [3]). For non-convex settings, recent works [4,5] also give convergence bounds that require $T \propto O(d)$ to converge (see Theorems 1,2,3,4 in [4] and Theorems 4.3, 5.2, Corollaries 4.6, 5.5 in [5]).
>
> As we have commented in our original submission (below Corollary 4.3), existing bounds for the parameter-free adaptive gradient method, D-adapted AdaGrad [6], are also similar to ours, and require $T \propto O(d)$ or even $T \propto O(d^2)$ to converge.
>
>
>
> >**Q2.** Corollary 4.4 and Corollary 5.2 state similar convergence rates for AdaGrad++ and Adam++. However, as mentioned by the author “AdaGrad++ as we found it consistently underperforms compared to Adam and Adam++”.
>
> **A2.**
> We would like to emphasize that our convergence rate guarantees are upper bounds characterizing the performance of the proposed algorithms in the worst case. In optimization literature, it is common that various algorithms share similar convergence rate guarantees in theory but perform differently in practice.
>
> For example, in literature, AdaGrad and Adam also have similar convergence rate bounds (see [4]: Theorems 1 and 3 are for AdaGrad, Theorems 2 and 4 are for Adam). However, It is common folklore that Adam performs better than AdaGrad in deep learning.
>
> Therefore, it is common to find that various algorithms, despite having similar theoretical convergence rate guarantees, perform differently in practice.

---

> ### Author Response · Authors · 2024-11-21
>
> >**Q3.** & **Q4.** The discrepancy between theorem proof and practical algorithm. Theorem 5.1 (Corollary 5.2) requires $\lambda \in (0,1)$, which actually disables the momentum update of Adam very quickly, seeing line-6 and line-7. Moreover, “Case 2” of the proposed Adam++ algorihtm requries the max operation $\max_{t’ \leq t } (\mathbf{v}_{t’})$.
>
> **A3.** & **A4.** We would like to clarify that our theoretical and practical settings both follow the standard in the literature. The max operation $\max_{t’ \leq t } (\mathbf{v}_{t’})$ and the decaying momentum parameter $\beta_{1t} = \beta_1\cdot \lambda^{t-1}$ follows the original AMSGrad paper [3] to ensure convergence guarantee. We would like to provide a bit of the background here:
>
> In 2018, a well-known work [3] constructed certain convex objective functions and pointed out that Adam may fail to converge when optimizing these convex objective functions. To fix this issue, [3] proposed the AMSGrad algorithm, which implements both
>
> (a) the max operation $\max_{t’ \leq t } (\mathbf{v}_{t’})$ (see Algorithm 2 in [3]), and
>
> (b) the decaying momentum parameter $\beta_{1t} = \beta_1\cdot \lambda^{t-1}$ (see Algorithm 2 and Corollary 1 in [3]).
>
> With these modifications, the authors of [3] established convergence guarantees of AMSGrad when $T \gg d$.
>
> Despite the issue of Adam pointed out by [3] in theory, Adam is still one of the most popular optimizers. And empirically, people find Adam outperforms AMSGrad in deep learning [7].
>
> Following this background above, we decided to follow [3] and consider the variant of Adam++ with (a) the max operation over $v_t$ and (b) the decaying momentum parameter $\beta_{1t} = \beta_1\cdot \lambda^{t-1}$ in our theoretical analysis.
>
> However, in experimental demonstration, we feel that it would be more convincing if we follow the convention and test the performance of Adam++ without modifications (a) and (b), as modifications (a) and (b) are mainly for the purpose of establishing convergence guarantees.
>
> Therefore, the ‘discrepancy’ between the theoretical proof and the practical algorithm is consistent with established literature. In fact, it also helps demonstrate that our proposed algorithms are robust to minor modifications in practical implementations.
>
>
>
> >**Q5.** Considering a parameter-free (or learning-rate-free) optimizer, it is expected to perform well on a spectrum of optimization tasks across different datasets, network architectures, and training parameters such as batch size, number of epochs, etc. I recognize that it is hard to comprehensively verify the method and lacks a uniform criterion of comparison. But universal adaptivity matters for LR-tuning-free methods.
>
> However, the conducted experiments in this work are insufficient to demonstrate the adaptivity compared with previous work. For example, in terms of the image classification tasks, Table 5 of [1] employed 5 diverse network architectures and 12 different datasets, while this work employed 2 diverse network architectures (ResNet* and VGG) and 1 dataset. Same for the language tasks, seeing Table 4 of [1]. The small-scale experiment cannot support the adaptivity of the proposed methods.
>
> **A5.** Thanks for your suggestions. We are working on adding more experiments and will update the paper when the results are ready. We would like to point out that our work also presents experiment setups that are not covered in [1]. For example, we have presented the results on GPT2.
>
>
> >**Q6.** I believe a golden choice of training settings is: Adam+LR(1e-3)+Cosine LR decay, which could be used as the baseline instead of a constant learning rate, since decreasing LR w.r.t. Iterations is well supported by many theoretical results. Table 1 in this paper demonstrates that the default choice of training method, Adam+LR(1e-3)+Cosine LR decay, outperforms the proposed methods.
>
> **A6.** First, we would like to clarify that one should not expect parameter-free methods to beat well-tuned Adam. The advantage of our algorithm (and other parameter-free algorithms) is not to beat Adam, but to *minimize the tuning effort*. As shown in Table 1, our proposed method outperforms the parameter-free baselines convincingly.
>
> We would also like to clarify that we cannot always use “Adam+LR(1e-3)+Cosine LR decay” as a golden standard. In our experiments, we have observed that on CIFAR-10, when training VGG16, ‘Adam+LR(1e-4)+Cosine LR decay’ works well, while ‘Adam+LR(5e-4)+Cosine LR decay’ or ‘Adam+LR(1e-3)+Cosine LR decay’ do not converge. We have added a set of preliminary experiment results in Appendix E in the revised paper. These results demonstrate the necessity for Adam to tune the initial learning rate, and demonstrate the advantage of our proposed parameter-free methods.

---

> ### Author Response · Authors · 2024-11-21
>
> >**Q7.** The main modification compared with the previous method is introducing $\sqrt{d}$
>  into the learning rate, however, this modification is not well supported intuitively or theoretically.
>
>
> **A7.** We believe that the reviewer may have overlooked the difference between our algorithm and DoG in [1]. Our main modification is not to introduce $\sqrt{d}$. Please note that the denominator in AdaGrad++ is different from DoG, and this is the key – AdaGrad++ applies adaptive learning rates entry-wisely, while DoG does not.
>
> Specifically, for DoG, the update rule is
>
> $ \mathbf{x}\_{t+1} =  \mathbf{x}\_{t} - \eta_t  \mathbf{g}_t, $ with $ \eta_t=\\frac{\\max\_{i\leq t} \\| \\mathbf{x}_0- \\mathbf{x}_i\\|_2}{\\sqrt{\\sum\_{i=1}^t \\| \\mathbf{g}_i \\|_2^2}}. $
>
> Clearly, this is exactly a variant of SGD with a specific choice of the learning rates $\eta_t$, and, as we have mentioned, the same learning rate $\eta_t$ is applied to all the entries to perform SGD update.
>
> In comparison, AdaGrad++ implements entry-wise adaptive learning rates:
>
> $\mathbf{x}\_{t+1} = \mathbf{x}\_{t} -  \frac{\eta_t}{\sqrt{\sum_{i=1}^t\mathbf{x}_i^2} + \delta}\cdot \mathbf{g}_t$ with  $   \eta_t = d^{-1/2} \\max\_{i\leq t} \\| \\mathbf{x}_0- \\mathbf{x}_i\\|_2, $
>
> where the division by $\sqrt{\sum_{i=1}^t\mathbf{x}_i^2} + \delta$ is performed entry-wisely. Clearly, the implementation of the term $\\max\_{i\leq t} \\| \\mathbf{x}_0- \\mathbf{x}_i\\|_2$ is motivated by (Ivgi et al., 2023), but the proposed algorithm is significantly different. Please also note that Adam++ is even more different from DoG.
>
> The reason for introducing the factor $\sqrt{d}$ is also that our proposed methods apply adaptive learning rates entry-wisely. Instead of directly using the total distance accumulating all entries, it is more reasonable to use the “mean square displacement” $d^{-1/2} \\| \\mathbf{x}_0- \\mathbf{x}_i\\|_2$.
>
>
> >**Q8.** Regarding the image classification task, Adam++ is employed, however, AdamW++ (Case 2) is employed for language tasks. Do you have any particular reasons for the switching&tuning operations? I also noticed two missing settings, i.e., AdamW++ (Case 1) and constant learning rate, in language tasks.
>
> **A8.** We observed that Adam++ and AdamW++ are comparable in image classification.
> For training large language models, AdamW is more widely used [7,8,9,10]. We also observe that AdamW++ outperforms Adam++ in training large language models. Therefore we present the results of AdamW++ for language tasks.
>
> ---
> ---
> **Reference**
>
> [1] Maor Ivgi, Oliver Hinder, and Yair Carmon. "Dog is sgd’s best friend: A parameter-free dynamic step size schedule". ICML, 2023.
>
> [2] Konstantin Mishchenko, and Aaron Defazio. "Prodigy: An expeditiously adaptive parameter-free learner". ICML, 2024.
>
> [3] Sashank J. Reddi, Satyen Kale, and Sanjiv Kumar. "On the Convergence of Adam and Beyond". ICLR, 2018.
>
> [4] Alexandre Défossez, Leon Bottou, Francis Bach, and Nicolas Usunier. "A Simple Convergence Proof of Adam and Adagrad". TMLR, 2024.
>
> [5] Dongruo Zhou, Jinghui Chen, Yuan Cao, Ziyan Yang, and Quanquan Gu. "On the Convergence of Adaptive Gradient Methods for Nonconvex Optimization". TMLR, 2024.
>
> [6] Aaron Defazio, and Konstantin Mishchenko. "Learning-rate-free learning by d-adaptation". ICML, 2023.
>
> [7] Sylvain Gugger, and Jeremy Howard. "Adamw and super-convergence is now the fastest way to train neural nets." last accessed, 2018.
>
> [8] Susan Zhang, Stephen Roller, Naman Goyal, Mikel Artetxe, Moya Chen, Shuohui Chen, Christopher Dewan et al. "Opt: Open pre-trained transformer language models." arXiv preprint arXiv:2205.01068 (2022).
>
> [9] Abhimanyu Dubey, Abhinav Jauhri, Abhinav Pandey, Abhishek Kadian, Ahmad Al-Dahle, Aiesha Letman, Akhil Mathur et al. "The llama 3 herd of models." arXiv preprint arXiv:2407.21783 (2024).
>
> [10] Aixin Liu, Bei Feng, Bin Wang, Bingxuan Wang, Bo Liu, Chenggang Zhao, Chengqi Dengr et al. "Deepseek-v2: A strong, economical, and efficient mixture-of-experts language model." arXiv preprint arXiv:2405.04434 (2024).

---

> > ### Comment · Reviewer_C9og · 2024-11-22
> >
> > Thanks for the detailed feedback!
> >
> > I have carefully reviewed all the feedback, and I will maintain my score for the following reasons:\
> > 1). As highlighted, the conducted experiments do not sufficiently support the proposed method's effectiveness. 2). The theoretical analysis lacks insights that adequately explain the success or validity of a parameter-free approach

---

> > > ### Author Response · Authors · 2024-11-25
> > >
> > > Dear Reviewer,
> > >
> > > Following your suggestions, we have added a series of additional experimental results, including training additional network architectures (Vision Transformer, Wide ResNet-50-2, and DenseNet-121) on additional datasets (CIFAR-10, CIFAR-100, and SVHN). Please find the results in Appendix E.2 of the revised paper. We believe that these additional results address your concerns about the experiments.
> > >
> > > Regarding your comments about our theoretical analysis, we would like to reemphasize that our analysis provides exactly the desired results for parameter-free algorithms. Please note that the classic convergence guarantees of AdaGrad and Adam typically rely on assumptions about the relationships between learning rates and the Lipschitz constant of the objective function, as well as knowledge of the global minimizer $x^*$ or its function value $f(x^*)$. This indicates that Adam and AdaGrad require adjusting learning rates according to the objective function. In comparison, our theoretical guarantees for AdaGrad++ and Adam++ do not rely on such assumptions about learning rates and still provide similar convergence guarantees as classic AdaGrad and Adam. Therefore, our results clearly demonstrate that AdaGrad++ and Adam++ are parameter-free algorithms that can achieve comparable performance to AdaGrad and Adam.
> > >
> > > If you have any additional questions, please let us know. Thank you!
> > >
> > > Best regards,
> > >
> > > Authors

---

> ### Author Response · Authors · 2024-11-23
>
> Dear Reviewer,
>
> Thank you for your quick feedback.
>
> Re: 1). As highlighted, the conducted experiments do not sufficiently support the proposed method's effectiveness.
>
> We are currently running additional experiments, including more architectures and datasets as per your suggestion, and will update the results in the coming days.
>
> Re: 2). The theoretical analysis lacks insights that adequately explain the success or validity of a parameter-free approach.
>
> We believe you may have overlooked our theoretical analysis. It explicitly demonstrates that AdaGrad++ and Adam++ achieve the existing best-known guarantees for AdaGrad and Adam, without requiring learning rate tuning. These theoretical guarantees meet precisely the expectations for parameter-free optimization algorithms.
>
> Please let us know if you have any other questions or suggestions. We will do our best to address them. Thanks.

---

> ### Author Response · Authors · 2024-11-25
>
> Dear Reviewer,
>
> Please note that we have updated our paper once again with several additional experiments. You can find all the added experiments in Appendix E of the revised paper. Additionally, you can refer to our post, 'Summary of Major Additional Experiments', for an overview of some major results. We are confident that these extensive new experimental results, along with our earlier responses to you, have addressed your concerns. We hope you can review them and let us know if you have any additional comments.
>
> Thank you!
>
> Best regards,
>
> Authors

---

> > ### Comment · Reviewer_C9og · 2024-11-27
> >
> > Thanks for the rebuttal!
> >
> > As mentioned, the theoretical analysis presented in the paper does not support the claims made in the paper and the statements in the rebuttal, such as achieving "convergence guarantees as classic AdaGrad and Adam". The assumptions used in the analysis appear unrealistic, and the derived results are not practically viable, such as a convergence speed that depends explicitly on the dimension $d$. Please refer to [1] for the latest advancements in Adam's convergence analysis.
> >
> > Regarding the experimental validation, I believe the scope of the current experiments is inadequate to demonstrate the effectiveness of a parameter-free optimizer. The inclusion of one language model with a single language dataset, along with a few well-known image datasets such as CIFAR10 and CIFAR100, does not provide a comprehensive evaluation.
> >
> > I will keep my score.
> >
> > [1] Ahn, Kwangjun, and Ashok Cutkosky. "Adam with model exponential moving average is effective for nonconvex optimization." arXiv preprint arXiv:2405.18199 (2024).

---

> ### Author Response · Authors · 2024-11-29
>
> Thank you for your feedback.
>
> > Re: the theoretical analysis presented in the paper does not support the claims made in the paper and the statements in the rebuttal, such as achieving "convergence guarantees as classic AdaGrad and Adam". The assumptions used in the analysis appear unrealistic, and the derived results are not practically viable, such as a convergence speed that depends explicitly on the dimension d. Please refer to [1] for the latest advancements in Adam's convergence analysis.
>
> [1] Ahn, Kwangjun, and Ashok Cutkosky. "Adam with model exponential moving average is effective for nonconvex optimization." arXiv preprint arXiv:2405.18199 (2024).
>
> We respectfully disagree. The paper [1] you referenced addresses nonconvex optimization, where the objective function is nonconvex, and the convergence guarantee is to a stationary point (i.e., the gradient norm converges to zero). In sharp contrast, our work focuses on convex optimization, where the objective function is convex, and the convergence guarantee is to the global optimal point. These are fundamentally different settings and are not directly comparable. We choose to focus on the convex optimization setting because nearly all existing literature on parameter-free optimization, including DoG, D-Adaptation, and Prodigy, is in this setting. Aligning with this line of research allows for meaningful comparisons and continuity. That said, we plan to extend our analysis to the nonconvex optimization setting in future work.
>
> Additionally, the linear dependence on dd arises from our assumption of the $\ell_infty$ norm on the gradient, rather than the $\ell_2$​ norm assumption. We have clearly explained this in the rebuttal, and this assumption is consistent with previous work.
>
>
> > Re: Regarding the experimental validation, I believe the scope of the current experiments is inadequate to demonstrate the effectiveness of a parameter-free optimizer. The inclusion of one language model with a single language dataset, along with a few well-known image datasets such as CIFAR10 and CIFAR100, does not provide a comprehensive evaluation.
>
> During the rebuttal, we added experiments on Vision Transformer, DenseNet, and WideResNet using the CIFAR-10, CIFAR-100, SVHN datasets. Additionally, we are currently running experiments on mini-ImageNet and will provide updates as soon as the results are available.
> For language model experiments, we evaluated our algorithm on GPT-2 small (155M) and medium (355M) models, using the widely adopted OpenWebText dataset. To the best of our knowledge, few, if any, existing works on parameter-free optimization have conducted such large-scale experiments on GPT-2 models.
>
> We believe that our experimental validation is as comprehensive as, if not more so than, closely related works such as DoG, D-Adaptation, and Prodigy. We kindly remind the reviewers to carefully review our experimental results.

---

### Official Review · Reviewer_FuLJ · 2024-11-02

**Soundness:** 2
**Presentation:** 2
**Contribution:** 1
**Rating:** 3
**Confidence:** 5

**Summary:**

This paper proposes two simple parameter-free variants of AdaGrad and Adam, called AdaGrad++ and Adam++. The authors also prove that the proposed algorithms including AdaGrad++ and Adam++ can achieve comparable convergence rates to their counterparts, AdaGrad and Adam. Some experimental results are reported.

**Strengths:**

The paper is complete in format.

**Weaknesses:**

As shown in Algorithm 1, the main differences between AdaGrad++ and Adam are from (Ivgi et al., 2023). Therefore, the novelty of this paper is limited. The experimental results are not convincing. The comparisons with recent algorithms are missing.

**Questions:**

1.	What’s the difference between the parameter-free techniques used in the proposed algorithm and existing ones?
2.	The detailed discussions about the convergence rates of the proposed algorithms and recently proposed algorithms are missing.
3.	The authors should compare the proposed algorithms with more recently proposed algorithms.

---

> ### Author Response · Authors · 2024-11-21
>
> Thank you for your helpful comments and suggestions. Please find our response to your comments and questions below. Due to the relatively long response, we will address your questions in two separate responses.
>
> >**Q1.** “As shown in Algorithm 1, the main differences between AdaGrad++ and Adam++ are from (Ivgi et al., 2023). Therefore, the novelty of this paper is limited”, “What’s the difference between the parameter-free techniques used in the proposed algorithm and existing ones?”
>
> **A1.** It is true that our proposed algorithms are inspired by (Ivgi et al., 2023). However, we would like to clarify that our proposed method is novel. (Ivgi et al., 2023) only proposed a parameter-free variant of SGD, and did not cover parameter-free adaptive gradient methods.  Specifically, please note that DoG proposed in (Ivgi et al., 2023) is similar to AdaGrad-Norm, which is not a fully adaptive gradient method, in the sense that all entries are using the same learning rate. Similarly for DoG, the update rule is
>
> $ \mathbf{x}\_{t+1} =  \mathbf{x}\_{t} - \eta_t  \mathbf{g}_t, $ with $ \eta_t=\\frac{\\max\_{i\leq t} \\| \\mathbf{x}_0- \\mathbf{x}_i\\|_2}{\\sqrt{\\sum\_{i=1}^t \\| \\mathbf{g}_i \\|_2^2}}. $
>
> Clearly, this is exactly a variant of SGD with a specific choice of the learning rates $\eta_t$, and, as we have mentioned, the same learning rate $\eta_t$ is applied to all the entries to perform SGD update.
>
> In comparison, AdaGrad++ implements entry-wise adaptive learning rates:
>
> $\mathbf{x}\_{t+1} = \mathbf{x}\_{t} -  \frac{\eta_t}{\sqrt{\sum_{i=1}^t\mathbf{g}_i^2} + \delta}\cdot \mathbf{g}_t$ with  $   \eta_t = d^{-1/2} \\max\_{i\leq t} \\| \\mathbf{x}_0- \\mathbf{x}_i\\|_2, $
>
> where the division by $\sqrt{\sum_{i=1}^t\mathbf{g}_i^2} + \delta$ is performed entry-wisely. Clearly, the implementation of the term $\\max\_{i\leq t} \\| \\mathbf{x}_0- \\mathbf{x}_i\\|_2$ is motivated by (Ivgi et al., 2023), but the proposed algorithm is significantly different. Please also note that Adam++ is even more different from DoG.
>
> Compared with existing parameter-free adaptive gradient methods proposed in recent works (Defazio & Mishchenko, 2023; Mishchenko & Defazio, 2023; Defazio et al., 2024), we would like to point out the following differences:
>
> 1. Most of these existing parameter-free adaptive gradient methods are not backed up with theoretical guarantees. Please note that most of the existing works considered parameter-free variants of both SGD and adaptive gradient methods, and theoretical guarantees are only established for the SGD variants. An exception is that (Defazio & Mishchenko, 2023) gives a theoretical guarantee for D-Adapted AdaGrad (Theorem 4 in their paper). We have compared our result with the theoretical guarantee for D-Adapted AdaGrad in our original submission (please see below Corollary 4.3).
>
> 2. As you have commented, AdaGrad++ and Adam++ are very intuitive and natural parameter-free variants of adaptive gradient methods. This is exactly the advantage and strength of our algorithms. To the best of our knowledge, these algorithms, even though very “simple”, have never been proposed in any existing works. More importantly, we prove that AdaGrad++ and Adam+ are competitive in terms of both theoretical guarantees and practical experiments. Therefore, one of our contributions is to formally point out that AdaGrad++ and Adam++ work well in both theoretical and experimental analysis.
>
> We would also like to argue that, given the good performance of AdaGrad++ and Adam++ in experiments, their relatively simple form (compared to other parameter-free adaptive gradient methods) should be a strength of our work, not a weakness.

---

> ### Author Response · Authors · 2024-11-21
>
> >**Q2.** The detailed discussions about the convergence rates of the proposed algorithms and recently proposed algorithms are missing.
>
>
> **A2.** As we have discussed in in **A1.**, to our knowledge, most of the existing parameter-free adaptive gradient methods do not have theoretical guarantees. An exception is D-Adapted AdaGrad proposed by (Defazio & Mishchenko, 2023), and we have discussed and compared it with our result below Corollary 4.3 in our original submission.
>
> If there are any specific results with which you would like us to compare, please let us know.
>
>
> >**Q3.** “The experimental results are not convincing. The comparisons with recent algorithms are missing”, “The authors should compare the proposed algorithms with more recently proposed algorithms”
>
> **A3.**
> First of all, we would like to clarify that the goal of our work is to deliver parameter-free versions of AdaGrad and Adam to achieve similar performance as well-tuned AdaGrad and Adam. Our goal is not to beat the original algorithms, but is to save efforts in hyperparameter tuning.
>
> We would like to emphasize that this paper is focused on parameter-free adaptive gradient methods. To our knowledge, the most related variants for Adam are D-Adapt Adam and prodigy, and we have already provided comparisons with these algorithms. We believe our experiments have served our purpose to demonstrate that our proposed method can achieve comparable performance as well-tuned Adam.
>
> If you would like us to compare any other related algorithms, please let us know.

---

> ### Author Response · Authors · 2024-11-25
>
> Dear Reviewer,
>
> We have revised the paper and added a number of additional experimental results. We are confident that our response and revisions have addressed all your concerns. As the discussion period is ending soon, we hope you can review our response and revisions and let us know if you have any additional comments.
>
> Thank you!
>
> Best regards,
>
> Authors

---

> > ### Author Response · Authors · 2024-12-02
> >
> > Dear Reviewer,
> >
> > We have not heard back from you since your original review. As there is only one day left for you to give us feedback, we sincerely hope you could check our response and revision, which we believe have fully addressed your concerns. Thank you.
> >
> > Best regards,
> >
> > Authors

---

### Author Response · Authors · 2024-11-25
**Summary of Major Additional Experiments**

Dear Reviewers,

Thank you for your helpful and constructive comments. To address your comments, we have added a number of additional experiment results. Here, we would like to give a summary of the experiment results we have added in the revision. In the additional experiments, we have compared different algorithms (Adam, D-Adapt Adam, Prodigy, Adam++(Case 1), Adam++ (Case 2), AdaGrad, and AdaGrad++) in training various network architectures (DenseNet-121, Vision Transformer, and Wide ResNet-50-2) on various datasets (CIFAR-10, CIFAR-100, and SVHN).

The following two tables compare the best accuracy (%) of different algorithms in 100 and 200 epochs on the CIFAR-10 dataset respectively:

|CIFAR-10, best accuracy in 100 epochs | | | | | | | |
| --------------- | --------- | ------------ | ------- | -------------- | --------------- | ------- | --------- |
| Models          | Adam  | D-Adapt Adam | Prodigy | Adam++(Case 1) | Adam++ (Case 2) | AdaGrad | AdaGrad++ |
| DenseNet-121     | 86.9  | 67.07        | 72.47   | 87.1           | **87.87**       | 65.8    | 86.32     |
| Vision Transformer       | 76.0  | 73.7         | 70.0    | **80.14**      | 78.76           | 65.13   | 76.53     |
| Wide ResNet-50-2 | 87.02 | 77.32        | 79.64   | 76.27          | **87.17**       | 63.91   | 75.12     |


|CIFAR-10, best accuracy in 200 epochs | | | | | | | |
| --------------- | --------- | ------------ | ------- | -------------- | --------------- | ------- | --------- |
| Models          | Adam      | D-Adapt Adam | Prodigy | Adam++(Case 1) | Adam++ (Case 2) | AdaGrad | AdaGrad++ |
| DenseNet-121     | 89.2      | 77.22        | 79.66   | **89.59**      | 89.07    | 66.39   | 89.58     |
| Vision Transformer     | 78.74     | 75.78        | 73.55   | **81.66**      | 80.09     | 66.49   | 79.4      |
| Wide ResNet-50-2 | **89.34** | 85.51        | 85.08   | 78.42    | 88.86     | 64.54   | 77.92     |

The two tables below present a comparison of the best accuracy (%) achieved by different algorithms in 100 and 200 epochs on the CIFAR-100 dataset:

|CIFAR-100, best accuracy in 100 epochs | | | | | | | |
| --------------- | --------- | ------------ | ------- | -------------- | --------------- | ------- | --------- |
| Models          | Adam      | D-Adapt Adam | Prodigy | Adam++(Case 1) | Adam++ (Case 2) | AdaGrad | AdaGrad++ |
| DenseNet-121     | 60.51     | 31.4         | 47.38   | 60.83  | **61.54**       | 36.86   | 59.49     |
| Vision Transformer    | 49.22     | 47.92        | 46.39   | **52.78**      | 51.41           | 23.99   | 50.34     |
| Wide ResNet-50-2 | **60.94** | 39.05        | 54.24   | 43.28          | 57.24           | 35.22   | 42.07     |

|CIFAR-100, best accuracy in 200 epochs | | | | | | | |
| --------------- | --------- | ------------ | ------- | -------------- | --------------- | ------- | --------- |
| Models          | Adam      | D-Adapt Adam | Prodigy | Adam++(Case 1) | Adam++ (Case 2) | AdaGrad | AdaGrad++ |
| DenseNet-121     | 63.05     | 43.44        | 54.25   | **64.05**      | 62.9            | 37.56   | 63.75     |
| Vision Transformer   | 52.91     | 51.78        | 50.6    | **55.68**      | 53.58           | 25.2    | 53.32     |
| Wide ResNet-50-2 | **64.33** | 51.97        | 59.25   | 46.5           | 61.11           | 35.38   | 45.76     |

Finally, shown in the following tables is a comparison of the best accuracy (%) attained by different algorithms in 100 and 200 epochs on the SVHN dataset:

|SVHN, best accuracy in 100 epochs | | | | | | | |
| --------------- | --------- | ------------ | ------- | -------------- | --------------- | ------- | --------- |
| Models          | Adam      | D-Adapt Adam | Prodigy | Adam++(Case 1) | Adam++ (Case 2) | AdaGrad | AdaGrad++ |
| DenseNet-121     | **95.45** | 79.01        | 88.15   | 95.41   | 95.32    | 79.27   | 94.83     |
| Vision Transformer  | 77.17     | 19.59        | 64.23   | **88.93**  | 35.09     | 70.36   | 85.69     |
| Wide ResNet-50-2 | 95.21     | 87.64        | 93.66   | 93.01    | **95.57**  | 87.07   | 95.20     |

|SVHN, best accuracy in 200 epochs | | | | | | | |
| --------------- | --------- | ------------ | ------- | -------------- | --------------- | ------- | --------- |
| Models          | Adam      | D-Adapt Adam | Prodigy | Adam++(Case 1) | Adam++ (Case 2) | AdaGrad | AdaGrad++ |
| DenseNet-121    | 95.51     | 89.46        | 92.94   | 95.53          | 95.34           | 80.35   | **95.73** |
| Vision Transformer  | 82.34     | 19.59        | 76.11   | **90.07**      | 44.06           | 73.36   | 87.84     |
| Wide ResNet-50-2 | **95.75** | 93.71        | 95.16   | 93.63          | 95.63           | 87.59   | 95.59     |

We believe these additional results can fully address your (Reviewers C9og and c1YW) initial concerns about the insufficient experiments. If you have any further questions, please let us know and we will try our best to address them. Thank you!

Best regards,

Authors

---

### Meta-Review · Area_Chair_LaQg · 2024-12-10

**Metareview:**

This paper introduces two parameter-free optimizers as variants of AdaGrad and Adam. The proposed methods aim to simplify hyperparameter tuning while maintaining theoretical guarantees and practical performance. While the theoretical contributions were acknowledged, reviewers raised concerns about practical relevance, novelty, experimental results, and computational overhead.

### **Strengths:**
- AdaGrad++ and Adam++ achieve convergence guarantees comparable to their classic counterparts under convex assumptions.
- General direction is of interest: parameter-free optimizers reduce the need for hyperparameter tuning
- During the rebuttal, the authors significantly expanded experiments, including new datasets (CIFAR-100, SVHN) and architectures (Vision Transformers, Wide ResNet, DenseNet).


### **Weaknesses:**
- Limited novelty: the core ideas are closely tied to the DoG framework, with limited innovation in the update rules for AdaGrad++ and Adam++. Some reviewers questioned whether the modifications were intuitive or theoretically necessary.
- Practical algorithms diverge slightly from their theoretical versions (e.g., the max operation and adjusting momentum parameters).
- Initial experiments lacked breadth, focusing on small-scale datasets like CIFAR-10 and limited architectures like ResNet and VGG. This was however addressed to a large extent.

While the paper has merits in providing parameter-free adaptive gradient methods, the mixed reviews highlight concerns about its contribution compared to prior work. I do want to acknowledge the lack of discussion from some reviewers (despite my best attempts) but the paper is overall significantly below the acceptance bar so I do think another round of reviews is necessary. I am unfortunately not able to recommend acceptance at the moment but I do encourage the authors to improve the paper to resubmit to a later deadline.

**Additional Comments On Reviewer Discussion:**

Some reviewers did not engage in a discussion even after I sent direct emails to them.

However, the paper is significantly below the acceptance bar.

---

### Decision · Program_Chairs · 2025-01-22

Reject